# IMPROVED ORDER ANALYSIS AND DESIGN OF EXPONENTIAL INTEGRATOR FOR DIFFUSION MODELS SAMPLING

## ABSTRACT

Efficient differential equation solvers have significantly reduced the sampling time of diffusion models (DMs) while retaining high sampling quality. Among these solvers, exponential integrators (EI) have gained prominence by demonstrating state-of-the-art performance. Surprisingly, we find existing high-order EI-based sampling algorithms rely on degenerate EI solvers, resulting in inferior error bounds and reduced accuracy in contrast to the theoretically anticipated results under optimal settings. This situation makes the sampling quality extremely vulnerable to seemingly innocuous design choices such as timestep schedules. For example, an inefficient timestep scheduler might necessitate twice the number of steps to achieve a quality comparable to that obtained through carefully optimized timesteps. To address this issue, we reevaluate the design of high-order differential solvers for DMs. Built upon well-established numerical framework from Hochbruck & Ostermann (2005a) and extensive experiments, we reveal that the degeneration of existing high-order EI solvers can be attributed to the absence of essential order conditions. By reformulating the differential equations in DMs and capitalizing on the theory of exponential integrators Weiner (1992), we introduce EI solvers that fulfill all order conditions for diffusion sampling, which we designate as Refined Exponential Solver (RES). Utilizing these improved solvers, RES exhibits more favorable error bounds theoretically and achieves superior sampling efficiency and stability in practical applications. For instance, a simple switch from the single-step DPM-Solver++ to our order-satisfied RES when Number of Function Evaluations (NFE) = 9, results in a reduction of numerical defects by $25.2\%$ and FID improvement of $25.4\%$ (16.77 vs 12.51) on a pre-trained ImageNet model.

## 1 INTRODUCTION

Diffusion models (DMs) (Ho et al., 2020; Song et al., 2021b) have recently garnered significant interest as powerful and expressive generative models. They have demonstrated unprecedented success in text-to-image synthesis (Rombach et al., 2022; Saharia et al., 2022; Balaji et al., 2022) and extended their impact to other data modalities, such as 3D objects (Poole et al., 2022; Lin et al., 2022; Shue et al., 2022; Bautista et al., 2022), audio (Kong et al., 2020), time series (Tashiro et al., 2021; Biloš et al., 2022), and molecules (Wu et al., 2022; Qiao et al., 2022; Xu et al., 2022). However, DMs have slow generation due to their iterative noise removal, requiring potentially thousands of network function evaluations (NFEs) to transform Gaussian noise into clean data, compared to Generative adversarial networks (GANs) single evaluation for a batch of images.

Recently, a surge of research interest has been directed toward accelerating the sampling process in diffusion models. One strategy involves the distillation of deterministic generation. Despite requiring additional computational resources, this approach can reduce the number of NFEs to fewer than five (Salimans & Ho, 2022; Song et al., 2023; Meng et al., 2022; Liu et al., 2023). Nevertheless, distillation methods often depend on training-free sampling methods during the learning process and are only applicable to the specific model being distilled, thereby limiting their flexibility compared to training-free sampling methods (Luhman & Luhman, 2021). Another line of investigation is aimed at designing generic sampling techniques that can be readily applied to any pre-trained DMs. Specifically, these techniques leverage the connection between diffusion models and stochastic differential equations (SDEs), as well as the feasibility of deterministically drawing samples by solving the equivalent Ordinary Differential Equations (ODEs) for marginal probabilities, known as probability flow ODEs (Song et al., 2021b;a). Research in this domain focuses on the design

of efficient numerical solvers to expedite the process (Liu et al., 2022; Zhang & Chen, 2022; Lu et al., 2022a; Zhang et al., 2022; Karras et al., 2022; Lu et al., 2022b; Zhao et al., 2023). Despite the significant empirical acceleration, numerous methods that claim the same order of numerical error convergence rate demonstrate significant disparities in practical application. Furthermore, these approaches often incorporate a range of techniques and designs that may obscure the actual factors contributing to the acceleration, such as various thresholding (Dhariwal & Nichol, 2021; Saharia et al., 2022), and sampling time optimization (DeepFloyd, 2023; Karras et al., 2022). (See App A for more related works and discussions.)

In this study, we initiate by revisiting different diffusion probability flow ODE parameterizations, underlining the vital role of approximation accuracy for integrals in the ODE solution in various sampling algorithms. This directly impacts both the quality and speed of sampling (Sec 3.1). Through a change-of-variable process, we further unify the ODEs for the noise prediction model and data prediction into a singular, canonical semilinear ODE. Building on the framework, and drawing from the wisdom of well-established numerical analysis literature (Hochbruck & Ostermann, 2005a; 2010), we conduct a detailed order analysis of diffusion ODE with semilinear structure. This analysis uncovers critical limitations in widely used sampling algorithms, notably their non-compliance with the order conditions of exponential integrators (EI). This leads to inferior error bounds, compromised accuracy, and a lack of robustness compared to the theoretically projected outcomes under optimal settings (Sec 3.2).

In response, we introduce the Refined Exponential Solver (`RES`), which is built upon an order-satisfied scheme originating from Weiner (1992). `RES` offers more favorable error recursion both theoretically and empirically, thus ensuring superior sampling efficiency and robustness (Sec 3.3). Moreover, our single-step scheme can be seamlessly integrated to enhance multistep deterministic (Sec 3.4) and stochastic sampling algorithms (Sec 3.5), surpassing the performance of contemporary alternatives. Lastly, we undertake comprehensive experiments with various diffusion models to demonstrate the generalizability, superior efficiency, and enhanced stability of our approach, in comparison with existing works (Sec 4). In summary, our contributions are as follows:

- We revisit and reevaluate diffusion probability flow ODE parameterizations and reveal the source of approximation error.

- To minimize error, we propose a unified canonical semilinear ODE, identify the overlooked order condition in existing studies, and rectify this oversight for diffusion ODE by leveraging theoretically sound exponential integrator analysis (Hochbruck & Ostermann, 2005a).

- We introduce `RES`, a new diffusion sampling algorithm that provides improved error recursion, both theoretically and empirically. Extensive experiments are conducted to show its superior efficiency, enhanced stability, and its orthogonality to other sampling improvements. For example, it shows up to 40% acceleration by switching single-step DPM-Solver++ to single-step `RES` under a suboptimal time scheduling.

## 2 BACKGROUND

Given a data distribution of interest, denoted as $p_{data}(\boldsymbol{x})$ where $\boldsymbol{x} \in \mathcal{X}$, a diffusion model consists of a forward noising process that diffuses data point $\boldsymbol{x}$ into random noise and a backward denoising process that synthesize data via removing noise iteratively starting from random noise. We start our discussion with simple variance-exploding diffusion models (Song & Ermon, 2020), which can be generalized to other diffusion models under the unifying framework introduced by Karras et al. (2022). Concretely, the forward noising process defines a family of distributions $p(\boldsymbol{x}; \sigma(t))$ dependent on time $t$, which is obtained by adding *i.i.d.* Gaussian noise of standard deviation $\sigma(t)$ to noise-free data samples. We choose $\sigma(t)$ to be monotonically increasing with respect to time $t$, such as $\sigma(t) = t$.

To draw samples from diffusion models, a backward synthesis process is required to solve the following stochastic differential equation (SDE) (Zhang & Chen, 2021; Huang et al., 2021; Karras et al., 2022), starting from $\boldsymbol{x}(T) \sim \mathcal{N}(\boldsymbol{0}, \sigma(T)^2 \boldsymbol{I})$ for a large enough $T$:

$$\mathrm{d}\boldsymbol{x} = - \underbrace{\dot{\sigma}(t)\sigma(t)\nabla_{\boldsymbol{x}} \log p(\boldsymbol{x}; \sigma(t))\mathrm{d}t}_{\text{Probabilistic ODE}} - \underbrace{\beta_t \sigma(t)^2 \nabla_{\boldsymbol{x}} \log p(\boldsymbol{x}; \sigma(t))\mathrm{d}t + \sqrt{2\beta_t}\sigma(t)\mathrm{d}\omega_t}_{\text{Langevin process}}, \quad (1)$$

where $\nabla_{\boldsymbol{x}} \log p(\boldsymbol{x}; \sigma(t))$ is the score function (*i.e.*, gradient of log-probability), $\omega_t$ is the standard Wiener process, and $\beta_t$ is a hyperparameter that controls the stochasticity of the process. Eq (1)

**Table 1:** Different parametrizations of the probabilistic ODE.

| Parametrization | Location | Velocity | Time | Semi-linear |
|---|---|---|---|---|
| EDM / DEIS | $\boldsymbol{x}$ | $(\boldsymbol{x} - \boldsymbol{D}_\theta(\boldsymbol{x}, t))/t$ | $t := \sigma$ | No |
| logSNR | $\boldsymbol{x}$ | $-\boldsymbol{x} + \boldsymbol{D}_\theta(\boldsymbol{x}, e^{-\lambda_D})$ | $\lambda_{\boldsymbol{D}} := -\log \sigma$ | Yes |
| Negative logSNR | $\boldsymbol{y} := \boldsymbol{x}/e^{\lambda_\epsilon}$ | $-\boldsymbol{y} + \epsilon_\theta(e^{\lambda_\epsilon}\boldsymbol{y}, e^{\lambda_\epsilon})$ | $\lambda_\epsilon := \log \sigma$ | Yes |

reduces to deterministic probabilistic flow ODE (Song et al., 2021b) when $\beta = 0$. Some popular methods, such as variance-preserving diffusion models, introduce an additional scale schedule $s(t)$ and consider $\boldsymbol{x} = s(t)\hat{\boldsymbol{x}}$ to be a scaled version of the original, non-scaled data $\hat{\boldsymbol{x}}$. Though the introduction of scale schedule $s(t)$ will result in a different backward process, we can undo the scaling and reduce their sampling SDEs to Eq (1). Note that we focus on non-scaling diffusion models in this paper for simplicity and leave the extension to other diffusion models in App B. The score function $\nabla_{\boldsymbol{x}} \log p(\boldsymbol{x}; \sigma(t))$ of noised data distribution at a particular noise level $\sigma$ can be learned via a denoising score matching objective (Vincent, 2011):

$$L_\sigma(\theta) = \mathbb{E}_{\boldsymbol{x} \sim p_{data}, \epsilon \sim \mathcal{N}(0, \boldsymbol{I})}[\|D_\theta(\boldsymbol{x} + \sigma\epsilon, \sigma) - \boldsymbol{x}\|_2^2], \tag{2}$$

where $D_\theta : \mathcal{X} \times \mathcal{R} \to \mathcal{X}$ is a time-conditioned neural network that tries to denoise the noisy sample. In the ideal case, having the perfect denoiser $D_\theta$ is equivalent to having the score function thanks to Tweedie's formula (Efron, 2011):

$$D_\theta(\boldsymbol{x}, \sigma) = \boldsymbol{x} + \sigma^2 \nabla_{\boldsymbol{x}} \log p(\boldsymbol{x}; \sigma). \tag{3}$$

For convenience, we also introduce the noise prediction model, which tries to predict $\epsilon$ in Eq (2):

$$\epsilon_\theta(\boldsymbol{x}, \sigma) := \frac{\boldsymbol{x} - D_\theta(\boldsymbol{x}, \sigma)}{\sigma}. \tag{4}$$

## 3 REFINED EXPONENTIAL SOLVER RES

Even with the same trained diffusion model, different solvers of Eq (1) will lead to samplers of drastically different efficiency and quality. In this section, we present a unified formulation to the various probability flow ODE being considered (Sec 3.1), analyze the numerical approximation errors for general single-step methods (Sec 3.2), and find the optimal set of coefficients that satisfies the desired order conditions and minimizes numerical error (Sec 3.3). We then extend the improvement to the stochastic setting (Sec 3.4) and multistep sampling methods (Sec 3.5).

### 3.1 BETTER PARAMETERIZATIONS FOR PROBABILITY FLOW ODE

The probability flow ODE in diffusion models in Eq (1) is first order, and thus we can draw some analogies between $\boldsymbol{x}$ and "location", $t$ and "time", and $-\dot{\sigma}(t)\sigma(t)\nabla_{\boldsymbol{x}} \log p(\boldsymbol{x}, \sigma(t))$ as "velocity". In general, an ODE is easier to solve if the "velocity" term has smaller first-order derivatives (Press et al., 2007; Lipman et al., 2022). Here, we list several different parametrizations for these quantities. While the exact solutions to these ODEs are the same, the numerical solutions can differ dramatically.

**EDM** In EDM (Karras et al., 2022), the location term is on the data space $\boldsymbol{x}$, the probability flow ODE in Eq (1) with $\beta_t = 0$ simplifies to the following equation:

$$\mathrm{d}\boldsymbol{x} = \frac{\boldsymbol{x} - \boldsymbol{D}_\theta(\boldsymbol{x}, \sigma(t))}{\sigma(t)} \dot{\sigma}(t) \mathrm{d}t. \tag{5}$$

To solve Eq (5), Karras et al. (2022) interpret the term $\frac{\boldsymbol{x} - \boldsymbol{D}_\theta(\boldsymbol{x}, \sigma(t))}{\sigma_t} \dot{\sigma}(t)$ as a black box function and choose $\sigma(t) = t$, and then apply a standard second-order Heun ODE solver.

**DEIS** Similarly, one varaint of DEIS (Zhang & Chen, 2022) uses the noise prediction model $\epsilon_\theta$ to parametrize the ODE:

$$\mathrm{d}\boldsymbol{x} = \epsilon_\theta(\boldsymbol{x}, \sigma)\mathrm{d}\sigma. \tag{6}$$

It is not hard to see that this is equivalent to the EDM one if we set $\sigma$ to $t$.

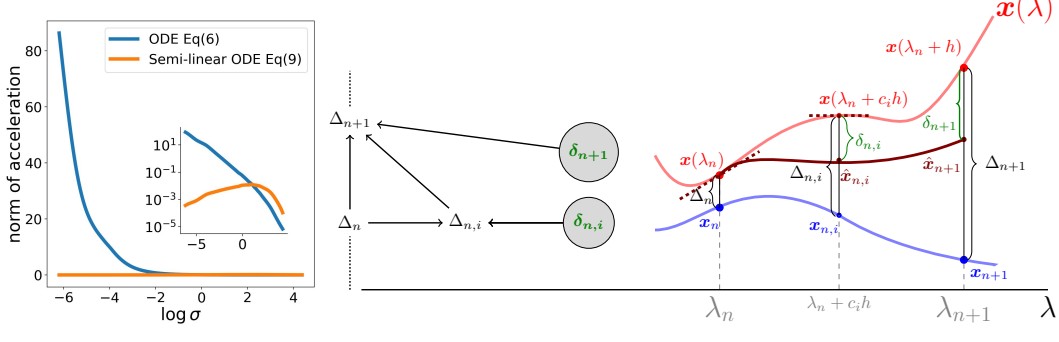

**(a)** Acceleration vs ODE        **(b)** Diagram of error recursion.

**Figure 1:** Fig 1a. Along the trajectory of an exact solution to the probability flow ODE, semi-linear ODE with logarithm transformation parametrization on noise level $\sigma$ has smaller "acceleration", *i.e.,* the time-derivative of "velocity". The curves are averaged over $512$ solutions on the pre-trained ImageNet model (Karras et al., 2022). This implies that semi-linear ODEs may incur lower discretization errors by approximating underlying networks with a constant or linear profile during the solving of ODEs. Fig 1b. The auxiliary trajectory $\hat{x}$ starts from exact solution $x$ and is built upon numerical integration proposed in Eq (11). The diagram depicts that numerical defects $\Delta_{n+1}$ between $x(\lambda_{n+1})$ and numerical solution $x_{n+1}$ is a result of accumulated intermediate defects $\{\delta_n, \delta_{n,i}\}$ in each step, which is the discrepancy between the auxiliary trajectory and the exact solution trajectory.

**logSNR**    We explore an alternative probability flow ODE parameterization based on the denoising model $D_\theta$. We consider $\lambda_D(t) := -\log \sigma(t)$, which can be treated as the log signal-to-noise ratio (logSNR, (Kingma et al., 2021)) between coefficients of the signal $x(0)$ and noise $\epsilon$. With $\mathrm{d}\lambda_D(t) = \frac{-\dot{\sigma}(t)}{\sigma(t)}\mathrm{d}t$, Eq (5) can be reformulated as follows:

$$\frac{\mathrm{d}x}{\mathrm{d}\lambda_D} = -x + g_D(x, \lambda_D), \quad g_D(x, \lambda_D) := D_\theta(x, e^{-\lambda_D}). \tag{7}$$

**Negative logSNR**    Alternatively, we can restructure Eq (6) by employing a change-of-variables approach with $y(t) := \frac{x(t)}{\sigma(t)}, \lambda_\epsilon(t) := \log \sigma(t)$ for the noise prediction model $\epsilon_\theta$:

$$\frac{\mathrm{d}y}{\mathrm{d}\lambda_\epsilon} = -y + g_\epsilon(y, \lambda_\epsilon), \quad g_\epsilon(y, \lambda_\epsilon) := \epsilon_\theta(e^{\lambda_\epsilon}y, e^\lambda_\epsilon), \tag{8}$$

where $\lambda_\epsilon$ is the negative logSNR. We observe that Eq (7) and (8) possess highly similar ODE structures. In fact, they are both *semilinear parabolic problems* in the numerical analysis literature (Hochbruck & Ostermann, 2005b), *i.e.*, the velocity term is a linear function of location plus a non-linear function of location ($g_D$ or $g_\epsilon$). This suggests that ODEs with either epsilon prediction or data prediction models can be addressed within a *unified* framework. We summarize these parametrizations in Tab 1.

A notable advantage of the logSNR and negative logSNR parametrizations is that their velocity terms along exact solution trajectory, such as $-x(\lambda_D) + g_D(x(\lambda_D), \lambda_D)$, are smoother than the velocity term in the ODE for EDM/DEIS. We illustrate this in Fig 1a, which shows that the norm of the derivative of the EDM velocity term grows rapidly as $\sigma \to 0$. To mitigate the challenges associated with solving ODEs that exhibit sharper "acceleration", the timestep scheduling employed in prevalent samplers (like DDIM and EDM) typically allocates more steps at lower noise levels.

For the rest of the section, we focus on logSNR (Eq (7)); theoretical conclusions and practical algorithms can be easily transferred to negative logSNR (Eq (8)). For notational simplicity, we substitute $\lambda$ for $\lambda_D$ and $g$ for $g_D$, so our ODE in Eq (7) becomes:

$$\frac{\mathrm{d}x}{\mathrm{d}\lambda} = -x + g(x, \lambda), \tag{9}$$

Suppose that our ODE solver advances for a step size of $h$ from $x(\lambda)$, then the exact solution $x(\lambda + h)$ to this ODE, as given by the *variation-of-constants* formula, is represented as follows:

$$x(\lambda + h) = e^{-h}x(\lambda) + \int_0^h e^{(\tau - h)}g(x(\lambda + \tau), \lambda + \tau)\mathrm{d}\tau. \tag{10}$$

However, the computation of this exact solution confronts two significant challenges: the intractability of the integral and the inaccessible integrand, which involves the evaluation of the denoiser on the exact solution $\boldsymbol{x}(\lambda + \tau)$. To approximate the intractable integration in Eq (10), we leverage numerical analysis and methods from Hochbruck & Ostermann (2005a) to derive order conditions and update scheme for single-step methods.

### 3.2 SINGLE-STEP NUMERICAL SCHEMES AND DEFECT ANALYSIS

To approximate the exact solution in Eq (10), the numerical scheme of explicit single-step methods, characterized by $s$-stages, can be described as follows:

$$\boldsymbol{x}_{n+1} = e^{-h_n}\boldsymbol{x}_n + h_n \sum_{i=1}^{s} b_i(-h_n)\boldsymbol{d}_{n,i}, \tag{11a}$$

$$\boldsymbol{x}_{n,i} = e^{-c_i h_n}\boldsymbol{x}_n + h_n \sum_{j=1}^{i-1} a_{ij}(-h_n)\boldsymbol{d}_{n,j}, \quad \boldsymbol{d}_{n,i} := g(\boldsymbol{x}_{n,i}, \lambda_n + c_i h_n), \tag{11b}$$

where $h_n := \lambda_{n+1} - \lambda_n$ denotes the step size from state $\boldsymbol{x}_n$ to $\boldsymbol{x}_{n+1}$, $\boldsymbol{x}_{n,i}$ denotes the numerical solution at the $i$-th stage with timestep $\lambda_n + c_i h_n$, and $\boldsymbol{d}_{n,i}$ is the function evaluation at the $i$-th stage. Notably, $\boldsymbol{x}_n$ denotes the numerical solution at time $\lambda_n$, which deviates from the exact solution $\boldsymbol{x}(\lambda_n)$; similarly, the numerical solution $\boldsymbol{x}_{n,i}$ at stage $i$ deviates from the exact $\boldsymbol{x}(\lambda_n + c_i h_n)$. The coefficients $a_{ij}, b_i, c_j$ in Eq (11) can be compactly represented using Butcher tableaus Tab 2 (Press et al., 2007).

**Table 2:** *Left:* Tableau form of $\{a_{ij}, b_i, c_j\}$ for numerical scheme in Eq (11). For all $1 \leq j \leq i \leq s$, $a_{ij} = 0$. *Middle:* For Euler method, $c_1 = 0, b_1 = 1$. *Right:* For Heun's method, $c_1 = 0, c_2 = 1, a_{21} = 1, b_1 = b_2 = 0.5$. Unlike our RES exponential integrator, the explicit Euler and Heun solvers do not have the additional exponential coefficients (*e.g.*, $e^{-h_n}$) in front of $\boldsymbol{x}_n$ (see Eq (11)). These coefficients are not reflected in the Butcher tableau.

$$
\begin{array}{c|cccc}
c_1 & & & & \\
c_2 & a_{21} & & & \\
\vdots & \vdots & \ddots & & \\
c_s & a_{s1} & \cdots & a_{s,s-1} & \\
\hline
& b_1 & \cdots & b_{s-1} & b_s
\end{array}
\qquad
\begin{array}{c|c}
0 & \\
\hline
& 1
\end{array}
\qquad
\begin{array}{c|cc}
0 & & \\
1 & 1 & \\
\hline
& 0.5 & 0.5
\end{array}
$$

To bound the numerical defect $\boldsymbol{x}_n - \boldsymbol{x}(\lambda_n)$, we first decompose it into the summation of, $\boldsymbol{x}_n - \hat{\boldsymbol{x}}_n$ and $\hat{\boldsymbol{x}}_n - \boldsymbol{x}(\lambda_n)$. Here, the auxiliary trajectory $\hat{\boldsymbol{x}}_n$ is obtained by substituding $\boldsymbol{d}_{n,i}$ in Eq (11) with $f(\lambda) := g(\boldsymbol{x}(\lambda), \lambda)$. Specifically, we define *intermediate numerical defects* $\delta_n, \delta_{n,i}$ as

$$\delta_{n,i} := \boldsymbol{x}(\lambda_n + c_i h_n) - \hat{\boldsymbol{x}}_{n,i}, \quad \hat{\boldsymbol{x}}_{n,i} = e^{-c_i h_n}\boldsymbol{x}(\lambda_n) - h_n \sum_{j=1}^{i-1} a_{ij}(-h_n)f(\lambda_n + c_j h_n), \tag{12a}$$

$$\delta_{n+1} := \boldsymbol{x}(\lambda_{n+1}) - \hat{\boldsymbol{x}}_{n+1}, \quad \hat{\boldsymbol{x}}_{n+1} = e^{-h_n}\boldsymbol{x}(\lambda_n) - h_n \sum_{i=1}^{s} b_i(-h_n)f(\lambda_n + c_i h_n). \tag{12b}$$

Intuitively, defects $\delta_{n,i}, \delta_{n+1}$ are caused by approximating the intractable integration in Eq (10) by finite summation in Eq (11). Let $\Delta_n := \boldsymbol{x}_n - \boldsymbol{x}(\lambda_n)$, $\Delta_{n,i} := \boldsymbol{x}_{n,i} - \boldsymbol{x}(\lambda_n + c_i h_n)$, we can derive error recursion with the help of $\delta_{n,i}, \delta_{n+1}$, such that

$$\Delta_{n+1} = e^{-h_n}\Delta_n + h\sum_{i=1}^{s} b_i(-h_n)(g(\boldsymbol{x}_{n,i}, \lambda_n + c_i h_n) - f(\lambda_n + c_i h_n)) - \delta_{n+1}, \tag{13a}$$

$$\Delta_{n,i} = e^{-c_i h_n}\Delta_n + h\sum_{j=1}^{i-1} a_{ij}(-h_n)(g(\boldsymbol{x}_{nj}, \lambda_n + c_j h_n) - f(\lambda_n + c_j h_n)) - \delta_{n,i}. \tag{13b}$$

Eq (13) reveals that numerical defects $\Delta_{n+1}, \Delta_{n,i}$ are influenced by the emergent intermediate defects $\delta_{n+1}, \delta_{n,i}$ and defects $\Delta_n$, which are themselves modulated by previous intermediate defects. Moreover, the discrepancy $g(\boldsymbol{x}_{n,i}, \lambda_n + c_i h_n) - f(\lambda_n + c_i h_n)$ is contingent upon $\Delta_{n,i}$. As error recursion illustrated in Fig 1b, the origin of final numerical defects lies in the defects $\delta_{n+1}, \delta_{n,i}$ in each step. Consequently, to curtail the ultimate defect, it becomes paramount to constrain $\delta_{n+1}, \delta_{n,i}$. To establish bounds for $\Delta_n$ using the above error recursion, the following lemma provides an expansion of intermediate defects, uncovering the association between $\delta_{n+1}, \delta_{n,i}$ and the derivative of $f(\lambda)$.

**Lemma 1** ((Hochbruck & Ostermann, 2005b)). *Let $\phi_j(-h_n) = \frac{1}{h_n^j} \int_0^{h_n} e^{\tau - h_n} \frac{\tau^{j-1}}{(j-1)!} d\tau$, the order-$q$ Taylor expansion of the intermediate defects $\delta_{n,i}, \delta_{n+1}$ can be formulated as follows:*

$$\delta_{n,i} = \sum_{j=1}^q h_n^j \psi_{j,i}(-h_n) f^{(j-1)}(\lambda_n) + \mathcal{O}(h_n^q) \quad \delta_{n+1} = \sum_{j=1}^q h_n^j \psi_j(-h_n) f^{(j-1)}(\lambda_n) + \mathcal{O}(h_n^q), \quad (14)$$

*where function $\psi_{j,i}, \psi_j$ for the coefficients of the $(j-1)$-th order derivative of $f(\lambda_n)$ are defined as*

$$\psi_{j,i}(-h_n) = \phi_j(-c_i h_n) c_i^j - \sum_{k=1}^{i-1} a_{ik}(-h_n) \frac{c_k^{j-1}}{(j-1)!}, \quad \psi_j(-h_n) = \phi_j(-h_n) - \sum_{k=1}^s b_k(-h_n) \frac{c_k^{j-1}}{(j-1)!}.$$

Lemma 1 furnishes an expansion expression for $\delta_{n,i}, \delta_{n+1}$, with each term contingent upon the derivative of $f$ and coefficients $\{\psi_j, \psi_{j,i}\}$. This implies that to minimize numerical defects ($\Delta_{n+1}$), an ODE parameterization that ensures the smoothest possible nonlinear function along the exact solution trajectory should be selected, thus substantiating our choice of the logarithmic transformation on the noise level. To this end, we can constrain numerical defects $\delta_{n+1}, \delta_{n,i}$ by minimizing the magnitude of $\{\psi_j, \psi_{j,i}\}$, forming the foundation of the order conditions for the single-step scheme.

### 3.3 ORDER CONDITIONS

With Eq (13) and Lemma 1, we can deduce the conditions necessary to achieve a numerical scheme of order $q$. We start with the case of a single-stage solver of order one.

**Theorem 1** (Error bound for solvers that satisfy the 1st-order condition). *When $\psi_1(-h_i) = 0$ is satisfied for $1 \leq i \leq n$, the error bound of first-order solver based on Eq (11)*

$$\|\boldsymbol{x}_n - \boldsymbol{x}(\lambda_n)\| \leq Ch \sup_{\lambda_{min} \leq \lambda \leq \lambda_{max}} \|f'(\lambda)\| \quad (15)$$

*holds for $h = \max_{1 \leq i \leq n} h_i$. The constant $C$ is independent of $n, h$.*

With Thm 1, $b_1(-h) = \phi_1(-h)$ and the numerical scheme reads

$$\boldsymbol{x}_{n+1} = e^{-h_n} \boldsymbol{x}_n + h_n \phi_1(-h) g(\boldsymbol{x}_n, \lambda_n), \quad (16)$$

which is known as exponential Euler (Hochbruck & Ostermann, 2010) or DDIM for diffusion model (Song et al., 2021a). Although our order analysis aligns with existing work (Lu et al., 2022a; Zhang & Chen, 2022), the distinction emerges in high-order methods.

**Theorem 2** (Error bound for solvers that satisfy the 1st- and 2nd-order conditions (Hochbruck & Ostermann, 2005a, Thm 4.7)). *When $\psi_1(-h_i) = \psi_2(-h_i) = \psi_{1,2}(-h_i) = 0$ is satisfied for $1 \leq i \leq n$, the error bound of second order solver based on Eq (11)*

$$\|\boldsymbol{x}_n - \boldsymbol{x}(\lambda_n)\| \leq Ch^2 (\sup_{\lambda_{min} \leq \lambda \leq \lambda_{max}} \|f'(\lambda)\| + \sup_{\lambda_{min} \leq \lambda \leq \lambda_{max}} \|f''(\lambda)\|) \quad (17)$$

*holds for $h = \max_{1 \leq i \leq n} h_i$. The constant $C$ is independent of $n, h$.*

To satisfy order conditions in Thm 2, the optimal Butcher tableau can be parameterized by $c_2$ as shown in Tab 3. It turns out the single-step scheme proposed in DPM-Solver++ (Lu et al., 2022b) can also be reformulated with a different Butcher tableau as shown in Tab 3 (See App B). Those two Butcher tableau was actually first proposed by Weiner (1992) for adaptive RK methods. Nevertheless, compared with Thm 2, DPM-Solver++ breaks the order condition $\psi_2(-h) = 0$ and introduces additional errors in solving ODEs (See App B) compared with Thm 2 theoretically. Practically, Hochbruck & Ostermann (2005a) demonstrated that numerical schemes satisfying order conditions yield enhanced performance in their low-dimensional synthetic problems. As evidenced in Sec 4, we verify our numerical scheme, which adheres to these order conditions, exhibits reduced numerical defects and consequently produces samples of higher quality from diffusion models.

**Proposition 1** (Informal). *Employing the Butcher tableau outlined in Tab 3, the single-step method Eq (11), characterized by three stages, functions as a third-order solver.*

---

**Algorithm 1** `RES` Second order Single Update Step with $c_2$

---

1: **procedure** SINGLEUPDATESTEP($\boldsymbol{x}_i, \sigma_i, \sigma_{i+1}$)
2: $\quad \lambda_{i+1}, \lambda_i \leftarrow -\log(\sigma_{i+1}), -\log(\sigma_i)$
3: $\quad h \leftarrow \lambda_{i+1} - \lambda_i$ $\qquad\qquad\qquad\qquad\qquad\qquad\qquad\qquad$ ▷ Step length
4: $\quad a_{21}, b_1, b_2 \leftarrow$ Tab 3 with $c_2$ $\qquad\qquad\qquad\qquad\qquad$ ▷ Runge Kutta coeffcients
5: $\quad (\boldsymbol{x}_{i,2}, \lambda_{i,2}) \leftarrow (e^{-c_2 h}\boldsymbol{x}_i + a_{21}hD_\theta(\boldsymbol{x}_i, \lambda_i), \lambda_i + c_2 h)$ $\qquad$ ▷ Additional evaluation point
6: $\quad \boldsymbol{x}_{i+1} \leftarrow e^{-h}\boldsymbol{x}_i + h(b_1 D_\theta(\boldsymbol{x}_i, \lambda_i) + b_2 D_\theta(\boldsymbol{x}_{i,2}, \lambda_{i,2}))$
7: $\quad$ **return** $\boldsymbol{x}_{i+1}$

---

**Table 3:** Butcher tableau comparison between `RES` and DPM-Solver++ (Lu et al., 2022b) for step size $h$, where $\phi_i := \phi_i(-h)$ is defined in Lemma 1, and we further define $\phi_{i,j} := \phi_{i,j}(-h) = \phi_i(-c_j h)$. †: $\gamma$ in third order `RES` needs to satisfy $2(\gamma c_2 + c_3) = 3(\gamma c_2^2 + c_3^3)$. The second order tableau is completely characterized by $c_2$, whereas the third order is parameterized by both $c_2$ and $c_3$.

|  | Second Order | Third Order$^\dagger$ | Order conditions |
|---|---|---|---|
| `RES` | $\begin{array}{c\|cc} 0 & & \\ c_2 & c_2\phi_{1,2} & \\ \hline 0 & \phi_1 - \frac{1}{c_2}\phi_2 & \frac{1}{c_2}\phi_2 \end{array}$ | $\begin{array}{c\|ccc} 0 & & & \\ c_2 & c_2\phi_{1,2} & & \\ c_3 & c_3\phi_{1,3} - a_{32} & \gamma c_2\phi_{2,2} + \frac{c_3^2}{c_2}\phi_{2,3} & \\ \hline & \phi_1 - b_2 - b_3 & \frac{\gamma}{\gamma c_2 + c_3}\phi_2 & \frac{1}{\gamma c_2 + c_3}\phi_2 \end{array}$ | Yes |
| DPM Solver++ | $\begin{array}{c\|cc} 0 & & \\ c_2 & c_2\phi_{1,2} & \\ \hline 0 & (1 - \frac{1}{2c_2})\phi_1 & \frac{1}{2c_2}\phi_1 \end{array}$ | $\begin{array}{c\|ccc} 0 & & & \\ c_2 & c_2\phi_{1,2} & & \\ c_3 & c_3\phi_{1,3} - a_{32} & \frac{c_3^2}{c_2}\phi_{2,3} & \\ \hline & \phi_1 - b_2 - b_3 & 0 & \frac{1}{c_3}\phi_2 \end{array}$ | Degenerate |

### 3.4 SINGLE-STEP UPDATE FOR DETERMINISTIC AND STOCHASTIC SAMPLER

With the single-step update scheme developed in Sec 3.3, we are prepared to develop a deterministic sampling algorithm by iteratively applying Eq (11) with the corresponding Butcher tableau. Inspired by the stochastic samplers proposed in Karras et al. (2022), which alternately execute a denoising backward step and a diffusion forward step, we can replace the original Heun-based single-step update with our improved single-step method. Concretely, one update step of our stochastic sampler consists of two substeps. The first substep simulates a forward noising scheme with a relatively small step size, while the subsequent substep executes a single-step ODE update with a larger step. We unify both samplers in Alg 2 where the hyperparameter $\eta$ determines the degree of stochasticity.

### 3.5 MULTI-STEP SCHEME

Instead of constructing a single-step multi-stage numerical scheme to approximate the intractable integration in Eq (10), we can employ a multi-step scheme that capitalizes on the function evaluations from previous steps. Analogous to the analysis in Sec 3.2 and conclusions in Thm 2, we can extend these results to a multi-step scheme, the details of which are provided in App E. Intuitively, multi-step approaches implement an update similar to Eq (11a). Instead of selecting intermediate states and incurring additional function evaluations, a multi-step method reuses function evaluation output from previous steps. In this work, our primary focus is on the multi-step predictor case. However, our framework and analysis can also be applied to derive multi-step corrector, which has been empirically proven to further improve sampling quality (Zhao et al., 2023; Li et al., 2023).

## 4 EXPERIMENTS

We further extend our experiments to address the following queries: (1) Do the deterministic numerical schemes of `RES` outperform existing methods in terms of reduced numerical defects and enhanced robustness? (2) Does a reduction in numerical defects translate to improved sampling quality? (3) Can our single-step numerical scheme enhance the performance of existing samplers?

**Numerical defects for deterministic sampler** First, we investigate the evolution of numerical defects $\|\boldsymbol{x}_N - \boldsymbol{x}(0)\|$ in $L1$ norm as the number of function evaluations (NFE) increases in pre-trained ImageNet diffusion models with default hyperparameters. Since the exact solution $\boldsymbol{x}(0)$ is unavailable, we approximate it using a 500-step RK4 solution, which exhibits negligible changes with additional steps. For a fair comparison, we evaluate the **second-order `RES`** against single-step

---

**Algorithm 2** RES Single-Step Sampler

1: **procedure** SINGLESAMPLER($D_\theta$, $\sigma_{i \in \{0,...,N\}}$, $\eta_{i \in \{0,...,N\}}$)
2:     **sample** $x_0 \sim \mathcal{N}(0, \sigma_0^2 I)$                    ▷ Generate initial sample at $\sigma_0$
3:     **for** $i \in \{0, \ldots, N-1\}$ **do**
4:         **sample** $\epsilon_i \sim \mathcal{N}(0, I)$
5:         $\bar{\sigma}_i \leftarrow \sigma_i + \eta_i \sigma_i$
6:         $\bar{x}_i \leftarrow x_i + \sqrt{\bar{\sigma}_i^2 - \sigma_i^2}\, \epsilon_i$                    ▷ Move from $\sigma_i$ to $\bar{\sigma}_i$ via adding noise
7:         $x_{i+1} \leftarrow$ SingleUpdateStep $(\bar{x}_i, \bar{\sigma}_i, \sigma_{i+1})$            ▷ Run single update step from $\bar{\sigma}_i$ to $\sigma_{i+1}$
8:     **return** $x_N$                    ▷ Return noise-free sample at $t_N$

---

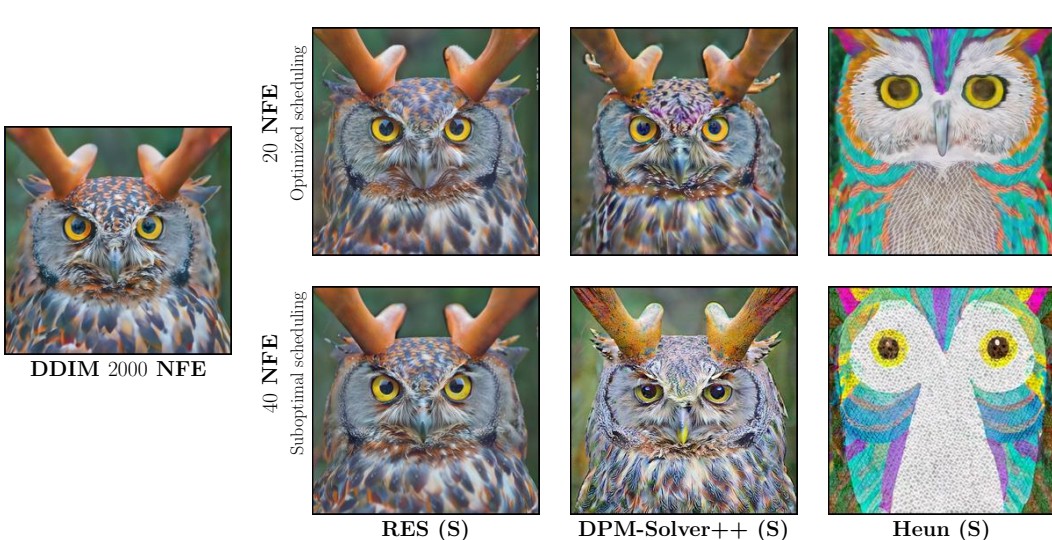

**Figure 2:** Importance of Order-Satisfied RK Tableau. RES showcases faster convergence with the same recommended time scheduling on (DeepFloyd, 2023) and better robustness against suboptimal scheduling compared with existing works. *Prompt: ultra close-up color photo portrait of rainbow owl with deer horns in the woods.*

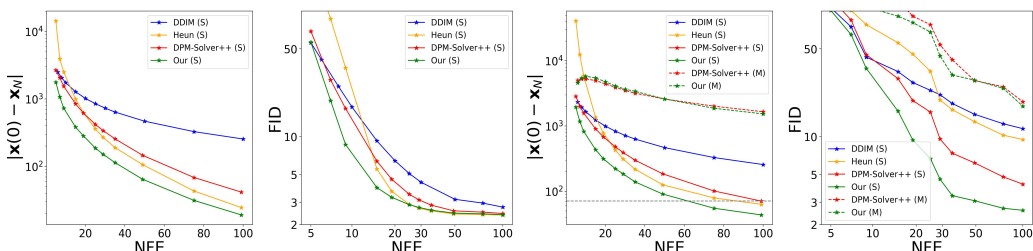

**Figure 3:** Comparison among various deterministic samplers on pretrained ImageNet diffusion model (Karras et al., 2022). (S) indicates single-step methods while (M) for multistep methods. (Left) Numerical defects and FID of various sampling algorithms *vs* the number of function evaluation (NFE) with recommended time scheduling (Karras et al., 2022). (Right) Numerical defects and FID *vs* NFEs with suboptimal time schedule. RES (S) shows better robustness against suboptimal-scheduling. Remarkably, with only 59 NFE, the single-step RES attains a numerical accuracy on par with the 99 NFE DPM-Solver++ (S).

DPM-Solver++(S)(Lu et al., 2022b) and Heun (Karras et al., 2022), both claimed to be second-order solvers. We also include first order DDIM (Song et al., 2021a) as a baseline. Our findings indicate that single-step RES exhibits significantly smaller numerical defects, consistent with our theoretical analysis. RES based on the noise prediction model (negative logSNR) surpasses the data prediction model (logSNR) in performance for guidance-free diffusion models; we provide a detailed presentation of the former here (More details in App F). Additionally, we compare single-step solvers

| NFE | 6 | 8 | 10 | 15 | 20 | 25 | 30 | 35 | 50 | 75 | 100 |
|---|---|---|---|---|---|---|---|---|---|---|---|
| DPM-Solver++ (M) | 14.87 | 8.02 | 5.46 | 3.52 | 2.98 | 2.73 | 2.60 | 2.52 | 2.42 | 2.40 | 2.38 |
| Our (M) | **14.32** | **7.44** | **5.11** | **3.23** | **2.54** | **2.41** | **2.38** | **2.35** | **2.34** | **2.33** | **2.33** |

**Table 4:** FID for DPM-Solver++ and `RES` musltistep across various NFEs on ImageNet Karras et al. (2022).

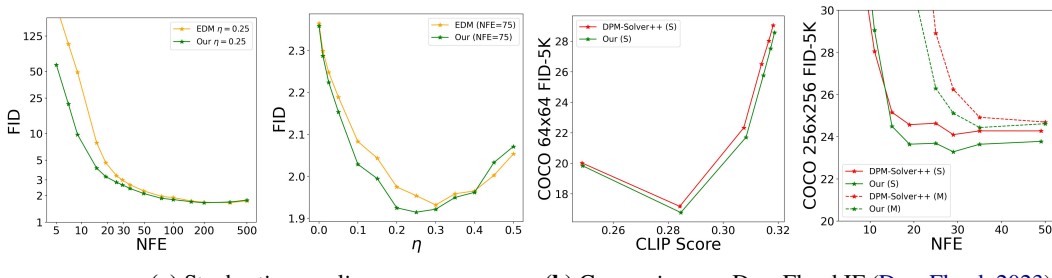

(a) Stochastic sampling.          (b) Comparison on DeepFloyd IF (DeepFloyd, 2023)

**Figure 4:** Fig 4a `RES` can also boost stochastic sampling by simply switching single-step update scheme from Heun (Karras et al., 2022) to ours under various NFEs and stochasticity. Fig 4b Our approach achieves superior performance in large-scale text-to-image diffusion models, excelling in both purely text-conditioned scenarios and super-resolution tasks conditioned on low-resolution images.

with multistep solvers, specifically including second-order multistep `RES` and DPM-Solver++ (M). We observe `RES-AB` consistently outperforms DPM-Solver++ (M) and both lead to smaller defects compared with single-step methods.

**Numerical defects and FID**    We further investigate the relationship between numerical defects and image sampling quality, using the Frechet Inception Distance (FID) (Heusel et al., 2017). As shown in Fig 3, smaller numerical defects generally lead to better FID scores as the number of function evaluations (NFE) increases for all algorithms. However, we also observe that the correlation may not be strictly positive and similar defects may lead to different FIDs. We observed a Pearson correlation coefficient of $0.956$ between these two metrics, which suggests that better FID scores are *strongly correlated* with smaller numerical defects. Notably, when NFE $= 9$, compared with single-step DPM-Solver++, the single step `RES` with noise prediction model (negative logSNR) reduce $52.75\%$ numerical defects, and $48.3\%$ improvement in FID (16.77 vs 8.66). For `RES` with data prediction model (logSNR), `RES` achieves $25.2\%$ numerical defects and $25.4\%$ FID improvement (16.77 vs 12.51).

**Time-scheduling robustness**    We also evaluate the performance of these solvers concerning varying timestep schedules. We observe that existing methods lack systematic strategies for optimal timestep selection, often relying on heuristic choices or extensive hyperparameter tuning. An ideal algorithm should demonstrate insensitivity to various scheduling approaches. Instead of using the recommended setting in EDM (Karras et al., 2022), we test these algorithms under a suboptimal choice with uniform step in $\sigma$. While all algorithms exhibit decreased performance under this setting, the single-step `RES` method outperforms the rest. Notably, single-step methods surpass multi-step methods in this scenario, which signifies the robustness of single-step `RES` and underscores the benefits of our principled single-step numerical scheme. Besides, to achieve a numerical accuracy comparable to that of the 99 NFE DPM-Solver++ (S), our single-step method requires only 59 NFE.

**Stochastic sampling with improved single-step algorithm**    Encouraged by the improvement of single-step sampling, we conduct experiments to show that this enhancement can boost stochastic samplers and achieve a favorable trade-off between sampling quality and speed. First, we investigate how FID is affected by NFE for a given $\eta$. Next, we sweep different values of $\eta$ under the same NFE. As illustrated in Fig 4a, we can accelerate the stochastic sampler compared with Heun-based EDM sampler (Karras et al., 2022).

**Cascaded text-to-image model**    Finally, we test algorithms on cascaded text-to-image models with DeepFloyd IF (DeepFloyd, 2023). We initially evaluate various sampling algorithms for $64 \times 64$ tasks

in Fig 4b. Furthermore, we compare the FID-CLIP score trade-off curves for various classifier-free guidance schemes under the same computational budget. Next, we incorporate a super-resolution diffusion model conditioned on generated images from the low-resolution model. In both scenarios, we find that our model consistently outperforms DPM-Solver++ in terms of quality.

## 5  CONCLUSION

In this work, we present the Refined Exponential Solver (`RES`), derived from a meticulous analysis of numerical defects in diffusion model probability flow ODEs and fulfilling all order conditions. `RES` enjoys superior error bounds theoretically and enhanced sampling efficiency in practice. Originally designed to boost single-step updates, we have also extended the improvements to multistep schemes and stochastic samplers, effectively enhancing their performance. We include more discussions and limitations in App D.

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

# A    RELATED WORKS AND DISCUSSION

To unlock and democratize the extraordinary generative potential of diffusion models, considerable research efforts have been dedicated to enhancing their efficiency and speeding up sampling. In addition to the works discussed in Sec 1, other methods have been explored to enhance the speed of diffusion models.

The authors of (Bao et al., 2022) optimize the backward Markovian process to approximate the non-Markovian forward process, yielding an analytic expression of optimal variance in the denoising process. Another strategy involves making the forward diffusion nonlinear and trainable (Zhang & Chen, 2021; Vargas et al., 2021; De Bortoli et al., 2021; Wang et al., 2021; Chen et al., 2021a), in the spirit of the Schrödinger bridge (Chen et al., 2021b). However, this approach incurs a considerable training overhead. Researchers have also explored modifying the backward stochastic process by incorporating more function evaluations and optimizing time scheduling (Kong & Ping, 2021; Watson et al., 2021). Nevertheless, such acceleration strategies struggle to generate high-quality samples with a limited number of discretization steps. An alternate approach for improving diffusion models involves designing non-isotropic diffusion models, such as Blurring diffusion models (Hoogeboom & Salimans, 2022; Rissanen et al., 2022) and the critically-damped Langevin diffusion models (Dockhorn et al., 2021). There are other fast sampling methods by construction diffusion models in latent space (Rombach et al., 2022; Vahdat et al., 2021). In addition, several works show GANs can be leveraged to accelerate diffusion models (Xiao et al., 2022; Wang et al., 2022).

More related are the diffusion model sampling algorithms inspired by semilinear ODEs solvers (Zhang & Chen, 2022; Lu et al., 2022a). Zhang & Chen (2022) introduced Diffusion Exponential Integrator Sampler (DEIS), whose fastest variant is based on an approximate exact solution, achieved by replacing the nonlinear function with polynomial exploration. However, DEIS was originally designed for fitting polynomials in the original $t$ space and is affected by noising scheduling $\sigma(t)$. On the other hand, Lu et al. (2022a) proposed DPM-Solver, which leverages a similar analysis of the exponential integrator to solve a semilinear ODE with time-varying coefficients. They further advanced DPM-Solver++ for the data prediction model and a sampling algorithm based on a multistep scheme, claiming improved sampling speed (Lu et al., 2022b). However, as we highlight in Sec 3.3, their numerical scheme does not meet all the necessary order conditions. The omission of these conditions could potentially worsen performance, especially with non-isotropic ODEs (Hochbruck & Ostermann, 2010).

# B    PROOF

In this section, we provide detailed derivations and proofs for the key theoretical results presented in the paper. Given that we have two semi-linear ODEs Eq (7) and (8) sharing similar structures, we primarily concentrate on the data prediction model, with the understanding that the results can be easily extended to the ODE based on the noise prediction model. Recall that we denote the denoiser network as $g$, and its evaluation along the exact solution $\boldsymbol{x}(\lambda)$ as $f(\lambda) := g(\boldsymbol{x}(\lambda), \lambda)$. Our analytical framework is heavily influenced by the foundational works of Hochbruck & Ostermann (2005a; 2010). We present self-contained and thorough derivations for the semi-linear ODE Eq (7), meticulously tailored for drawing samples from diffusion models.

## B.1    PROOF OF LEMMA 1

We first expand $f$ into a Taylor series with the remainder in integral form,

$$f(\lambda_n + \tau) = \sum_{j=1}^{q} \frac{\tau^{j-1}}{(j-1)!} f^{(j-1)}(\lambda_n) + \int_0^{\tau} \frac{(\tau - \nu)^{q-1}}{(q-1)!} f^{(q)}(\lambda_n + \nu) \mathrm{d}\nu \tag{18}$$

With Eq (18), we can rewrite exact ODE solution Eq (10) as

$$\boldsymbol{x}(\lambda_n + c_i h) = e^{-c_i h} \boldsymbol{x}(\lambda_n) + \sum_{j=1}^{q_i} (c_i h)^j \phi_j(-c_i h) f^{(j-1)}(\lambda_n) \tag{19}$$

$$+ \int_0^{c_i h} e^{-(c_i h - \tau)} \int_0^{\tau} \frac{(\tau - \nu)^{q_i - 1}}{(q_i - 1)!} f^{(q_i)}(\lambda_n + \nu) \mathrm{d}\nu \mathrm{d}\tau,$$

where function $\phi_j$ is defined as

$$\phi_j(-h) = \frac{1}{h^j} \int_0^h e^{\tau-h} \frac{\tau^{j-1}}{(j-1)!} \mathrm{d}\tau, \quad j \geq 1. \tag{20}$$

With integration by part, we arrive at the recursion formulation for $\phi$ function (Hochbruck & Oster-mann, 2005a) as

$$\phi_{k+1}(z) = \begin{cases} \frac{\phi_k(z) - 1/k!}{z} & k \geq 0, z \neq 0 \\ e^z & k = 0 \\ \frac{1}{k!}, & z = 0, k \leq 0 \end{cases}. \tag{21}$$

One the other hand, by integrating Eq (18) into the numerical scheme Eq (11), we can derive another expression for the exact solution. We begin with the exact solution at the intermediate time $\lambda_n + c_i h$. Plugging in Eq (18) into Eq (12a), we obtain

$$\boldsymbol{x}(\lambda_n + c_i h) = e^{-c_i h} \boldsymbol{x}(\lambda_n) + h \sum_{k=1}^{i-1} a_{ik}(-h) \sum_{j=1}^{q_i} \frac{(c_k h)^{j-1}}{(j-1)!} f^{(j-1)}(\lambda_n) \tag{22}$$

$$+ h \sum_{k=1}^{i-1} a_{ik}(-h) \int_0^{c_k h} \frac{(c_k h - \nu)^{q_i-1}}{(q_i - 1)!} f^{(q_i)}(\lambda_n + \nu) \mathrm{d}\nu \mathrm{d}\tau + \delta_{n,i}.$$

By comparing Eq (19) and Eq (22), we obtain the error term $\Delta_{ni}$ as

$$\delta_{n,i} = \sum_{j=1}^{q_i} h^j \psi_{j,i}(-h) f^{(j-1)}(\lambda_n) + \delta_{n,i}^{(q_i)} \tag{23}$$

$$\psi_{j,i}(-h) = \phi_j(-c_i h) c_i^j - \sum_{k=1}^{i-1} a_{ik}(-h) \frac{c_k^{j-1}}{(j-1)!}, \tag{24}$$

where $\delta_{n,i}^{(q_i)}$ denotes higher order terms that resulted from the truncation of the Taylor series.

Similarly, we can get the expression for $\delta_{n+1}$ if we plug Eq (18) into Eq (12b) and comparing the result against Eq (19)

$$\delta_{n+1} = \sum_{j=1}^{q} h^j \psi_j(-h) f^{(j-1)}(\lambda_n) + \delta_{n+1}^{(q)} \tag{25}$$

$$\psi_j(-h) = \phi_j(-h) - \sum_{k=1}^{s} b_k(-h) \frac{c_k^{j-1}}{(j-1)!} \tag{26}$$

where $\delta_{n+1}^{(q)}$ denotes higher order terms that resulted from the truncation of the Taylor series.

## C   PROOF OF ERROR BOUND

We first state several mild assumptions that are required for our theoretical bounds.

**Assumption 1.**   *For nonlinear function $g : \boldsymbol{R}^d \times \boldsymbol{R} \to \boldsymbol{R}^d$ considered in this work, e.g. $\epsilon_\theta$, $\boldsymbol{D}_\theta$, we assume there exists a real number $L(R)$ such that*

$$\|\boldsymbol{g}(\boldsymbol{x}_1, \lambda) - \boldsymbol{g}(\boldsymbol{x}_2, \lambda)\| \leq L \|\boldsymbol{x}_1 - \boldsymbol{x}_2\|. \tag{27}$$

*for all $\lambda_{\min} \leq \lambda \leq \lambda_{max}$ and $\max(\|\boldsymbol{x}_1 - \boldsymbol{x}(\lambda)\|, \|\boldsymbol{x}_2 - \boldsymbol{x}(\lambda)\|) \leq R$ where $\boldsymbol{x}(\lambda)$ is one ODE solution with nonlinear function $g$.*

The region where $\boldsymbol{x}_1, \boldsymbol{x}_2$ exists and is close to the exact solution is referred to as the strip along the exact ODE solution. For high-order methods, we introduce another assumption regarding the nonlinear function $g$:

**Assumption 2.**   *For ODE with nonlinear function $g : \boldsymbol{R}^d \times \boldsymbol{R} \to \boldsymbol{R}^d$ considered in this work, e.g. $\epsilon_\theta$, $\boldsymbol{D}_\theta$, $g$ is differentiable in a strip along the ODE exact solution $\boldsymbol{x}(\lambda)$. All occurring derivatives are uniformly bounded.*

By default, we assume that Thm 1 and 2 and Prop 1 satisfy above assumptions. First, we consider a scenario with a uniform step size, denoted as $h$. Subsequently, the obtained results are extended to accommodate non-uniform step sizes. This generalization involves loosening the error bounds' reliance on the step size from the uniform case, directing it instead towards the maximum step size evident in the non-uniform scenario. Indeed, the error boundaries delineated in Thm 1 and 2 manifest dependence on this longest step size.

First, we bound the truncated high-order term $\delta_{n,i}^{(q_i)}$. Based on Eq (19) and (22), $\delta_{n,i}^{(q_i)}$ can be bounded by

$$
\begin{aligned}
\|\delta_{n,i}^{(q_i)}\| \leq & \| \int_0^{c_i h} e^{-(c_i h - \tau)} \int_0^\tau \frac{(\tau - \nu)^{q_i - 1}}{(q_i - 1)!} f^{(q_i)}(\lambda_n + \nu) d\nu d\tau \| \\
& + \| h \sum_{k=1}^{i-1} a_{ik}(-h) \int_0^{c_k h} \frac{(c_k h - \nu)^{q_i - 1}}{(q_i - 1)!} f^{(q_i)}(\lambda_n + \nu) d\nu \| \\
\leq & \sup_{\lambda \in [\lambda_n, \lambda_n + c_i h]} \|f^{(q_i)}\| \left( \| \int_0^{c_i h} e^{-(c_i h - \tau)} \int_0^\tau \frac{(\tau - \nu)^{q_i - 1}}{(q_i - 1)!} d\nu d\tau \| \right. \\
& \left. + \| h \sum_{k=1}^{i-1} a_{ik}(-h) \int_0^{c_k h} \frac{(c_k h - \nu)^{q_i - 1}}{(q_i - 1)!} d\nu \| \right) \\
\leq & \sup_{\lambda \in [\lambda_n, \lambda_n + c_i h]} \|f^{(q_i)}\| C (c_i h)^{q_i + 1},
\end{aligned}
\tag{28}
$$

where $C$ is a sufficiently large constant. This is due to

$$
\| \int_0^{c_i h} e^{-(c_i h - \tau)} \int_0^\tau \frac{(\tau - \nu)^{q_i - 1}}{(q_i - 1)!} d\nu d\tau \| \leq \| \int_0^{c_i h} d\tau \| \| \int_0^{c_i h} \frac{(c_i h - \nu)^{q_i - 1}}{(q_i - 1)!} \| = \frac{(c_i h)^{q_i + 1}}{(q_i)!}
$$

$$
\| h \sum_{k=1}^{i-1} a_{ik}(-h) \int_0^{c_k h} \frac{(c_k h - \nu)^{q_i - 1}}{(q_i - 1)!} d\nu \| \leq \frac{\| \sum_{k=1}^{i-1} a_{ik}(-h) \|}{c_i} \frac{(c_i h)^{q_i + 1}}{(q_i)!}.
$$

We note above conclusion only holds when $\psi_{j,i}(-h) = 0$ is satisfied. Otherwise, defects of $h^j f^{(j-1)}(\lambda_n)$ will show up $\|\delta_{n,i}^{(q_i)}\|$ once $\psi_{j,i}(-h) \neq 0$.

Similarly, we can bound the accumulation of $\delta_i^{(q_i)}$ with

$$
\| \sum_{j=0}^{n-1} e^{-jh} \delta_{n-j}^{(q_i)} \| \leq C h^{q_i} \sup_{\lambda \in [\lambda_1, \lambda_n]} \|f^{(q_i)}\|
\tag{29}
$$

once $\phi_j = 0$ is satisfied. Indeed, the defect $\delta_n^{(q)}$ can be formulated

$$
\|\delta_n^{(q_i)}\| = \| \int_0^h e^{-(h - \tau)} \int_0^\tau \frac{(\tau - \nu)^{(q_i - 1)}}{(q_i - 1)!} f^{(q_i)}(\lambda_{n-1} + \nu) d\nu d\tau \|
\tag{30}
$$

$$
\leq \| \int_0^h e^{-(h - \tau)} d\tau \| \| \int_0^h \frac{(h - \nu)^{(q_i - 1)}}{(q_i - 1)!} d\nu \| \sup_{\lambda \in [\lambda_{n-1}, \lambda_n]} \|f^{(q_i)}\|
\tag{31}
$$

$$
= (1 - e^{-h}) \frac{h^{q_i}}{q_i!} \sup_{\lambda \in [\lambda_{n-1}, \lambda_n]} \|f^{(q_i)}\|
\tag{32}
$$

Therefore, we can bound

$$\|\sum_{j=0}^{n-1} e^{-jh}\delta_{n-j}^{(q_i)}\| \leq \sum_{j=0}^{n-1}\|e^{-jh}\delta_{n-j}^{(q_i)}\|$$

$$\leq \|\sum_{j=0}^{n-1} e^{-jh}\|(1-e^{-h})\frac{h^{q_i}}{q_i!}\sup_{\lambda\in[\lambda_1,\lambda_n]}\|f^{(q_i)}\|$$

$$\leq \frac{1}{1-e^{-h}}(1-e^{-h})\frac{h^{q_i}}{q_i!}\sup_{\lambda\in[\lambda_1,\lambda_n]}\|f^{(q_i)}\|$$

$$= \frac{h^{q_i}}{q_i!}\sup_{\lambda\in[\lambda_1,\lambda_n]}\|f^{(q_i)}\| \leq Ch^{q_i}\sup_{\lambda\in[\lambda_1,\lambda_n]}\|f^{(q_i)}\|,$$

with a sufficiently large $C$.

Second, we introduce Discrete Gronwall Inequality in Lemma 2, which we will use later.

**Lemma 2** (Discrete Gronwall Inequality). *Let $\langle\alpha_n\rangle$ and $\langle\beta_n\rangle$ be nonnegative sequences and c a nonnegative constant. If*

$$\alpha_n \leq c + \sum_{0\leq k<n}\beta_k\alpha_k,$$

*then*

$$\alpha_n \leq c\prod_{0\leq j<n}(1+\beta_j) \leq c\exp\big(\sum_{0\leq j<n}\beta_j\big) \quad for \quad n\geq 0. \tag{33}$$

### C.1 ORDER 1 ERROR BOUND FOR THM 1

With Lemma 1, we can formulate the error recursion for the $s=1$ case

$$\Delta_{n+1} = e^{-h}\Delta_n + h\phi_1(-h)(g(\boldsymbol{x}_n,t_n)-f(t_n)) - \delta_{n+1} \tag{34}$$

with defects $\delta_{n+1} = \delta_{n+1}^{(1)}$, the recursion gives

$$\Delta_n = h\sum_{j=0}^{n-1} e^{(-n-j-1)h}\phi_1(-h)(g(\boldsymbol{x}_j,t_j)-f(t_j)) - \sum_{j=0}^{n-1} e^{-jh}\delta_{n-j}. \tag{35}$$

Thanks to Assumption 1 and Eq (29), we can bound above $\Delta_n$ by

$$\|\Delta_n\| \leq \sum_{j=1}^{n-1} h\phi_1(-h)e^{-(n-j-1)h}L\|\Delta_j\| + Ch\sup_{\lambda\in[\lambda_n,\lambda_n+c_ih]}\|f^{(1)}\| \tag{36}$$

With Discrete Gronwall Inequality Lemma 2 and $\Delta_0 = 0$, we can show that

$$\|\Delta_n\| = \|\boldsymbol{x}_n - \boldsymbol{x}(\lambda_n)\| \leq Ch\sup_{\lambda_{\min}\leq\lambda\leq\lambda_{\max}}\|f'(\lambda)\|\exp\big(\prod_{j=1}^{n-1}hL\phi_1(-h)e^{-(n-j-1)h}\big) \tag{37}$$

$$\leq Ch\sup_{\lambda_{\min}\leq\lambda\leq\lambda_{\max}}\|f'(\lambda)\|\exp(L^n) \tag{38}$$

$$\leq C'h\sup_{\lambda_{\min}\leq\lambda\leq\lambda_{\max}}\|f'(\lambda)\| \tag{39}$$

with a large enough $C'$. This finishes the proof for Thm 1 if we upper bound Eq (39) by the largest step size in the non-uniform stepsize case. Thanks to order condition $\psi_1 = 0$ and $c_1 = 0$, the Butcher tableau follows

$$\begin{array}{c|c} 0 & \\ \hline & \phi_1(-h). \end{array} \tag{40}$$

### C.2 High order error bounds

We first start with the second order method. Since it is a second-stage algorithm, we have only one intermediate point $x_{n,1}$. Based on Eq (13) and (28) and Assumption 1, we can bound

$$\|\Delta_{n,2}\| \le C_1 \|\Delta_n\| + \|\delta_{n,2}^{(1)}\| \tag{41}$$

$$\le C_1 \|\Delta_n\| + C_2 h^2 \sup_{\lambda \in [\lambda_n, \lambda_n + c_2 h]]} \|f^{(1)}\|, \tag{42}$$

where $C_1, C_2$ are two constants. And we know $\Delta_{n,1} = \Delta_n$ for our explicit method due to $c_1 = 0$.

Similar to Eq (36), we can bound

$$\|\Delta_n\| \le h \sum_{j=0}^{n-1} e^{(-n-j-1)h} \phi_1(-h) C_3 \max\left(\|\Delta_j\|, \|\Delta_{j,1}\|\right) + \sum_{j=0}^{n-1} e^{-jh} \delta_{n-j}\|$$

$$\le h \sum_{j=0}^{n-1} e^{(-n-j-1)h} \phi_1(-h) C_4 \|\Delta_j\| + h \sum_{j=0}^{n-1} e^{(-n-j-1)h} \phi_1(-h) C_2 h^2 \sup_{\lambda \in [\lambda_{min}, \lambda_{max}]} \|f^{(1)}\|$$

$$+ C_5 h^2 \sup_{\lambda \in [\lambda_{min}, \lambda_{max}]} \|f^{(2)}\|,$$

where $C_4 = \max(1, C_1)$, and the last inequality is due to Eq (29). We note that $h \sum_{j=0}^{n-1} e^{(-n-j-1)h} \phi_1(-h) \le 1$. We can further simplify the upper bound of $\Delta_n$ by

$$\|\Delta_n\| \le h \sum_{j=0}^{n-1} e^{(-n-j-1)h} \phi_1(-h) C_4 \|\Delta_j\|$$

$$+ \max(C_2, C_5) h^2 \left( \sup_{\lambda \in [\lambda_{min}, \lambda_{max}]} \|f^{(2)}\| + \sup_{\lambda \in [\lambda_{min}, \lambda_{max}]} \|f^{(1)}\| \right).$$

Due to Lemma 2, we can upper bound it by

$$\|\Delta_n\| \le \max(C_2, C_5) h^2 \left( \sup_{\lambda \in [\lambda_{min}, \lambda_{max}]} \|f^{(2)}\| + \sup_{\lambda \in [\lambda_{min}, \lambda_{max}]} \|f^{(1)}\| \right) \tag{43}$$

$$\exp\left( \sum_{j=1}^{n-1} C_4 h \phi_1(-h) e^{-(n-j-1)h} \right)$$

$$\le C' h^2 \left( \sup_{\lambda \in [\lambda_{min}, \lambda_{max}]} \|f^{(2)}\| + \sup_{\lambda \in [\lambda_{min}, \lambda_{max}]} \|f^{(1)}\| \right) \tag{44}$$

for a large enough $C'$. This finishes the proof of Thm 2 if we upper bound Eq (39) by the largest step size in the non-uniform stepsize case.

Thanks to condition $\psi_1(-h) = \psi_2(-h) = \psi_{1,2}(-h) = 0$, the Butcher tableaus has to satisfy
$$b_1 + b_2 = \phi_1(-h)$$
$$b_2 c_2 = \phi_2(-h)$$
$$a_{21} = c_2 \phi_1(-c_2 h).$$
Therefore, the only solution for Butcher tableau is

$$\begin{array}{c|cc} 0 & & \\ c_2 & c_2 \phi_1(-c_2 h) & \\ \hline 0 & \phi_1(-h) - \frac{1}{c_2}\phi_2(-h) & \frac{1}{c_2}\phi_2(-h) \end{array}. \tag{45}$$

For third-order methods, numerical methods with three-stage methods have to satisfy (Hochbruck & Ostermann, 2005a, Sec 5.2)
$$\psi_1(-h) = 0$$
$$\psi_2(-h) = 0$$
$$\psi_{1,2}(-h) = 0$$
$$\psi_{1,3}(-h) = 0$$
$$\psi_3(-h) = 0$$
$$b_2 \psi_{2,2}(-h) + b_3 \psi_{2,3}(-h) = 0$$

---

**Algorithm 3** RES General $s$-stage Single-step Update with $\{c_s\}$

---

1: **procedure** SINGLEUPDATESTEP($\boldsymbol{x}_n, \sigma_n, \sigma_{n+1}$)
2:     $\lambda_{n+1}, \lambda_n \leftarrow -\log(\sigma_{n+1}), -\log(\sigma_n)$
3:     $h \leftarrow \lambda_{n+1} - \lambda_n$                                                        ▷ Step length
4:     $\{a_{ij}\}, \{b_j\} \leftarrow$ Butcher tableau with $\{c_i\}$       ▷ Single-step update coeffcients
5:     $(\boldsymbol{x}_{n,1}, \lambda_{n,1}) \leftarrow (\boldsymbol{x}_n, \lambda_n)$                      ▷ $c_1 = 0$ for explicit methods
6:     **for** $i$ in $2, \cdots, s$ **do**
7:         $(\boldsymbol{x}_{n,i}, \lambda_{n,i}) \leftarrow (e^{-c_i h}\boldsymbol{x}_n + h\sum_{j=1}^{i} a_{i,j}\boldsymbol{D}_\theta(\boldsymbol{x}_{n,j}, \lambda_n + c_j h), \lambda_n + c_i h)$
8:     $\boldsymbol{x}_{n+1} \leftarrow e^{-h}\boldsymbol{x}_n + h(\sum_{i=1}^{s} b_i \boldsymbol{D}_\theta(\boldsymbol{x}_{n,i}, \lambda_n + c_i h))$
9:     **return** $\boldsymbol{x}_{n+1}$

---

One two-parameter solution family for Butcher tableau satisfies other conditions and achieves third-order error bounds follows

$$
\begin{array}{c|cccc}
0 & & & & \\
c_2 & c_2\phi_{1,2} & & & \\
c_3 & c_3\phi_{1,3} - a_{32} & \frac{c_3^2}{c_2}\phi_{2,3} & & \\
\hline
& \phi_1 - b_2 - b_3 & 0 & \frac{1}{c_3}\phi_2 &
\end{array}, \tag{46}
$$

where $\gamma$ can be obtained by solving $2(\gamma c_2 + c_3) = 3(\gamma c_2^2 + c_3^3)$ once $c_2, c_3$ are given.

## D    DISCUSSIONS AND LIMITATIONS

Our work is also constrained by the following limitations. Our analysis reveals that the performance of different samplers is contingent on the trajectory of network evaluations along the exact solution, where a time-smooth trajectory facilitates the reduction of numerical error. However, existing works are still uncertain about which time transformation or time scheduling leads to the most favorable trajectory. The logarithmic transformation we've used was selected based on our empirical observations. In practical terms, training-free methods still require more than 10 network evaluations, making them significantly slower than GANs or distillation-based methods. We leave it for future work to explore how distillation training can benefit from these improved training-free methods.

## E    ALGORITHMS

For deterministic single-step and stochastic samplers, we have listed the unified algorithm in Alg 2. We have also listed the algorithm for single-step second order update in Alg 1. Furthermore, we list general high order algorithm in Alg 3.

### E.1    MULTISTEP ALGORITHM

The insights and analysis gained from a multi-stage single-step numerical scheme can be leveraged to develop multistep numerical methods. The crux of a multistep update step lies in its utilization of not only the function evaluation at the current state but also previous function evaluations.

To underscore the parallels between single-step and multistep methods, consider a single update step from timestamp $\lambda_n$ to $\lambda_{n+1} := \lambda_n + h$ with function evaluations $\{g(\boldsymbol{x}_{n,i}, \lambda_n + c_i h)\}_{i=1}^{r}$, where $c_1 = 0, c_i < 0$ for $1 < i \leq r$ and $\boldsymbol{x}_{n,i}$ are numerical results on timestamp $\lambda_n + c_i h$. The general multistep scheme can be expressed as follows:

$$
\boldsymbol{x}_{n+1} = e^{-h}\boldsymbol{x}_n + h[\sum_{i=1}^{r} b_i g(\boldsymbol{x}_{n,i}, \lambda_n + c_i h)] \tag{47}
$$

The present multistep method configuration bears many resemblances to the single-step method outlined in Sec 3.2. The distinction resides in the selection of $\{c_i\}_{i=2}^{r}$, which are chosen such that $\lambda_n + c_i h = \lambda_{n+1-i}$. Consequently, with $\boldsymbol{x}_{n,i}$ coinciding with $\boldsymbol{x}_{n+1-i}$, we can bypass the function evaluation cost, as $g(\boldsymbol{x}_{n,i}, \lambda_n + c_i h)$ is readily available due to the existing value of $g(\boldsymbol{x}_{n+1-i}, \lambda_{n+1-i})$.

From this viewpoint, the construction of auxiliary $\hat{\boldsymbol{x}}$ and intermediate numerical defects $\delta_n$ in Sec 3.2 remain applicable for multistep cases. Most notably, the error recursion in Eq (13a) and the expansion

---

**Algorithm 4** RES Second order Multistep Update Scheme

1: **procedure** MULTISTEPUPDATE($\boldsymbol{x}_n, \sigma_n, \sigma_{n+1}, \boldsymbol{D}_\theta(\boldsymbol{x}_{n-1}, \lambda_{n-1})$ )
2: $\quad \lambda_{n+1}, \lambda_n, \lambda_{n-1} \leftarrow -\log(\sigma_{n+1}), -\log(\sigma_n), -\log(\sigma_{n-1})$
3: $\quad h \leftarrow \lambda_{n+1} - \lambda_n$                ▷ Step length
4: $\quad c_2 \leftarrow \frac{\lambda_{n-1} - \lambda_n}{h}$
5: $\quad b_1, b_2 \leftarrow \phi_1(-h) - \phi_2(-h)/c_2, \phi_2(-h)/c_2$       ▷ Coeffcients
6: $\quad \boldsymbol{x}_{n+1} \leftarrow e^{-h}\boldsymbol{x}_n + h(b_1 D_\theta(\boldsymbol{x}_n, \lambda_n) + b_2 D_\theta(\boldsymbol{x}_{n-1}, \lambda_{n-1}))$
7: $\quad$ **return** $\boldsymbol{x}_{n+1}$

---

of $\delta_{n+1}$ in Lemma 1 persist in the multistep scenario. To minimize numerical defects, we also aim to reduce $\delta_n$. Therefore, the choice of $b_i$ from the Butcher tableaus developed for the single-step scheme can be employed for multistep methods with particular $c_i$ values. The second-order multistep algorithm is detailed in Alg 4.

## F   EXPERIMENT DETAILS

In this section, we provide necessary experiment details and extra experiments.

### F.1   REPRODUCTION OF FIG 1

**Fig 1a** We use the pre-trained ImageNet $64 \times 64$ class-conditioned diffusion model (Karras et al., 2022) to conduct the experiment. Due to the lack of ground truth solution trajectories, we use high-accuracy ODE solvers to approximate the solutions. Concretely, we employ RK4 with 500 steps (2000 NFE) to generate trajectories, with negligible alterations with more steps. To generate those trajectories, we randomly select initial random noise and labels. Curves in Fig 1a are averaged over 512 trajectories.

**Extra experiments** Besides the "acceleration" outlined in Fig 1a, we examine the evolution of $f(\lambda) := g(\boldsymbol{x}(\lambda), \lambda)$ along the solution trajectory for various nonlinear functions $g$. We note that smoother $f$ typically results in smaller numerical defects, as indicated in Lemma 1 and Thm 1 and 2. As depicted in Fig 5, the benefit of applying a logarithmic transformation to the noise level $\sigma$ is evident.

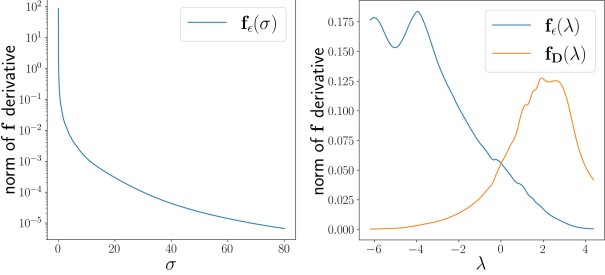

**Figure 5:** (Left) The evolution of nonlinear function evaluation for Eq (6) characterized by $f_\epsilon(\sigma) := \epsilon_\theta(\boldsymbol{x}(\sigma), \sigma)$. (Right) The evolution of nonlinear function evaluation for Eq (8), defined by $f_\epsilon(\lambda) := \epsilon_\theta(\boldsymbol{x}(\lambda), \lambda)$ and $f_D(\lambda) := D_\theta(\boldsymbol{x}(\lambda), \lambda)$. Implementing a logarithmic transformation on the noise level $\sigma$ results in smoother trajectories of nonlinear function evaluations along ODE solutions.

### F.2   REPRODUCTION OF FIG 3

We utilize the pre-trained ImageNet $64 \times 64$ class-conditioned diffusion model (Karras et al., 2022) for our experiment. For the numerical defect experiments, we approximate the ground truth solution $\boldsymbol{x}(0)$ using 500 steps RK4 (2000 NFE). We generate $50,000$ images with randomized labels to calculate numerical defects and FIDs. The EDM, as suggested by Karras et al. (2022), follows a time schedule as given in Eq (48),

$$t_i = (\sigma_{\max}^{\frac{1}{\rho}} + \frac{i}{N-1}(\sigma_{\min}^{\frac{1}{\rho}} - \sigma_{\max}^{\frac{1}{\rho}}))^\rho, \quad (48)$$

where $\rho$ is a hyperparameter that controls the timestamp spacing. Our findings reveal that the $\epsilon_\theta$-based RES outperforms the $\boldsymbol{D}_\theta$-based RES. By default, we adhere to the recommended $\rho = 7$ for our

| NFE | DDIM (S) | Heun (S) | DPM-Solver++ (S) | Our (S) | DPM-Solver++ (M) | Our (M) |
|---|---|---|---|---|---|---|
| 6 | 2515 | $1.446 \times 10^4$* | 2681* | 1774* | 1507 | 2602 |
| 8 | 2063 | 3943* | 2060* | 1072* | 1062 | 1895 |
| 10 | 1747 | 2501* | 1544* | 729.1* | 740.2 | 1195 |
| 15 | 1280 | 992.7 | 845.7 | 382.6 | 369.9 | 463.3 |
| 20 | 1020 | 623.9* | 617.8* | 283.6* | 227.6 | 258.7 |
| 25 | 852.7 | 360.4 | 420.7 | 187.6 | 153.2 | 167.8 |
| 30 | 731.4 | 270.4* | 339.4* | 151.1* | 114 | 117.1 |
| 35 | 640.1 | 190.2 | 256.5 | 113.8 | 87.17 | 85.13 |
| 50 | 468.4 | 106.3* | 146* | 64.43* | 46.06 | 40.59 |
| 75 | 328.1 | 43.23 | 68.56 | 31.35 | 22.82 | 18.28 |
| 100 | 254.6 | 24.56* | 41.63* | 19.08* | 14.58 | 10.65 |

**Table 5:** Numerical defects $|\boldsymbol{x}(0) - \boldsymbol{x}_N|$ with different NFEs and recommended time scheduling for Fig 3 (Left). * indicates the number is produced by one less NFE.

| NFE | DDIM (S) | Heun (S) | DPM-Solver++ (S) | Our (S) | DPM-Solver++ (M) | Our (M) |
|---|---|---|---|---|---|---|
| 6 | 41.09 | 248.3* | 68.97* | 56.46* | 14.87 | 14.32 |
| 8 | 25.12 | 86.88* | 28.33* | 19.4* | 8.024 | 7.44 |
| 10 | 17.26 | 35.32* | 16.77* | 8.665* | 5.456 | 5.115 |
| 15 | 9.284 | 5.544 | 6.402 | 3.944 | 3.525 | 3.23 |
| 20 | 6.446 | 3.682* | 4.593* | 3.298* | 2.975 | 2.542 |
| 25 | 5.101 | 2.899 | 3.493 | 2.885 | 2.726 | 2.412 |
| 30 | 4.348 | 2.705* | 3.151* | 2.742* | 2.602 | 2.377 |
| 35 | 3.712 | 2.571 | 2.86 | 2.607 | 2.523 | 2.346 |
| 50 | 3.18 | 2.431* | 2.575* | 2.471* | 2.425 | 2.336 |
| 75 | 2.968 | 2.406 | 2.514 | 2.436 | 2.401 | 2.335 |
| 100 | 2.755 | 2.381 | 2.453 | 2.401 | 2.377 | 2.333 |

**Table 6:** FID with different NFEs and recommended time scheduling for Fig 3 (Mid). * indicates the number is produced by one less NFE.

experiments. To test the robustness through the suboptimal time rescheduling experiment, we use $\rho = 1$, where the timestamps are uniformly distributed in $\sigma$. For all experiments, we test the sampling algorithm with NFE set to $6, 8, 10, 15, 20, 25, 30, 35, 50, 75, 100$. Similar to EDM, an additional denoising step is included in the final stage. We note that for second-order single-step methods, the total NFE may not align perfectly with the NFE. In such instances, we always take one less step, for example, second-order methods only use 5 NFEs when we expect them to utilize 6 NFEs. We use the official code from Karras et al. (2022) to calculate FID score. We include quantitative results in Tab 5 to 7.

| NFE | DDIM (S) | Heun (S) | DPM-Solver++ (S) | Our (S) | DPM-Solver++ (M) | Our (M) |
|---|---|---|---|---|---|---|
| 6 | 2339 | $4.116 \times 10^4$* | 2850* | 1957* | 5034 | 4615 |
| 8 | 1948 | $1.249 \times 10^4$* | 2004* | 1177* | 5281 | 5430 |
| 10 | 1664 | 5901* | 1578* | 827.9* | 5296 | 5800 |
| 15 | 1241 | 1362 | 907.2 | 435.6 | 4975 | 5469 |
| 20 | 996.9 | 780.2* | 681* | 314.8* | 4445 | 4779 |
| 25 | 834.9 | 428 | 483.1 | 222.6 | 3923 | 4116 |
| 30 | 718.5 | 312.7* | 392.9* | 184.8* | 3513 | 3675 |
| 35 | 632.6 | 218.3 | 300.5 | 139.6 | 3191 | 3409 |
| 50 | 467.5 | 126.1* | 185.6* | 90.87* | 2597 | 2607 |
| 75 | 329.4 | 79.22 | 101.2 | 55.01 | 2019 | 1876 |
| 100 | 256.8 | 63.15* | 70.82* | 43.44* | 1658 | 1537 |

**Table 7:** Numerical defects $|\boldsymbol{x}(0) - \boldsymbol{x}_N|$ with different NFEs and suboptimal time scheduling for Fig 3 (Right). * indicates the number is produced by one less NFE.

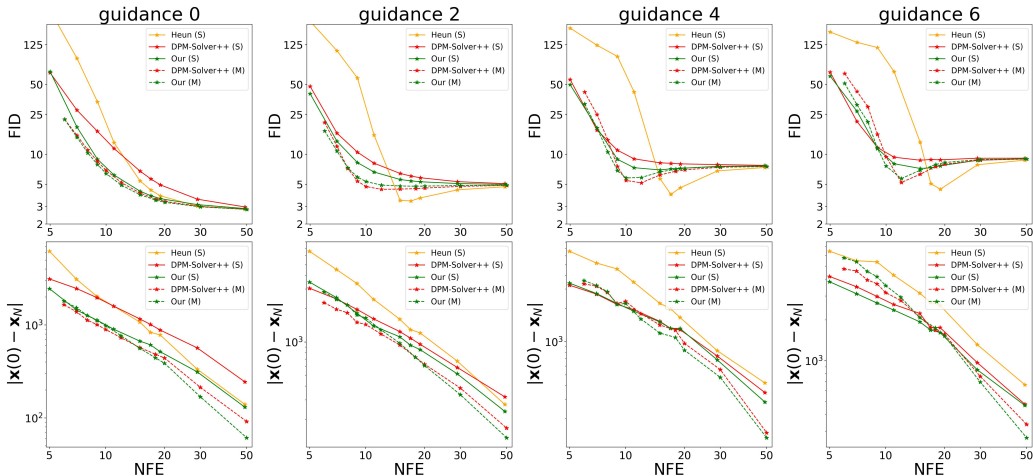

**Figure 6:** Comparison on DDPM-like classifier-guided diffusion models (Dhariwal & Nichol, 2021) with different guidance.

### F.3 MORE EXPERIMENTS ON CLASSIFIER-GUIDED DMs

We further investigate RES diffusion models trained using different schemes, including DDPM-like diffusion models with classifier-based guidance (Dhariwal & Nichol, 2021). It is important to note that we employ dynamic thresholding (Saharia et al., 2022) for all experiments to help alleviate pixel over-saturation issues. To ensure a fair comparison, all algorithms differ solely in their numerical update schemes. As depicted in Fig 6, our previous observations remain valid for guided diffusion models trained with different techniques, and RES converges faster than baselines.

Concretely, we utilize pre-trained diffusion models equipped with a noise data classifier from Dhariwal & Nichol (2021). These diffusion models are trained on discrete time analogous to DDPM. By employing an additional noise data classifier $p(\boldsymbol{c}|\boldsymbol{x})$, we can derive a new diffusion model by incorporating the gradient of the classifier into the diffusion model, resulting in the following denoiser:

$$\hat{\boldsymbol{D}}(\boldsymbol{x}, \sigma|\boldsymbol{c}) = \boldsymbol{x} + \sigma^2 \nabla_{\boldsymbol{x}} \log p(\boldsymbol{x}|\boldsymbol{c}; \sigma) \tag{49}$$

$$= \boldsymbol{x} + \sigma^2 \nabla_{\boldsymbol{x}} \log p(\boldsymbol{x}; \sigma) + \sigma^2 \nabla_{\boldsymbol{x}} \log p(\boldsymbol{c}|\boldsymbol{x}; \sigma), \tag{50}$$

where $p(\boldsymbol{c}|\boldsymbol{x}; \sigma)$ denotes a classifier for the label $\boldsymbol{c}$ on noise data. In practice, researchers have observed that increasing the weight on the classifier can enhance performance, as expressed by:

$$\hat{\boldsymbol{D}}(\boldsymbol{x}, \sigma|\boldsymbol{c}) \approx \hat{\boldsymbol{D}}_\theta(\boldsymbol{x}, \sigma|\boldsymbol{c}) + \omega \sigma^2 \nabla_{\boldsymbol{x}} \log p(\boldsymbol{y}|\boldsymbol{x}; \sigma), \tag{51}$$

where $\omega$ can be adjusted to values greater than 1. In our experiment, we utilize the pre-trained classifier detailed in Dhariwal & Nichol (2021). We approximate the ground truth $\boldsymbol{x}(0)$ using 500 steps of the RK4 method. In addition, we apply dynamic thresholding (Saharia et al., 2022) to all methods evaluated in this experiment to ensure a fair comparison. We conduct the sampling tests with NFE values of $6, 7, 8, 9, 10, 12, 15, 18, 20, 30, 50$. It should be noted that for guided diffusion models, our approach based on $\boldsymbol{D}_\theta$ proves superior. We use the official code from Dhariwal & Nichol (2021) to calculate FID score.

### F.4 REPRODUCTION OF FIG 4A

This experiment is designed to demonstrate how an enhanced single-step sampler can augment stochastic samplers' performance. We utilize the pre-trained ImageNet $64 \times 64$ class-conditioned diffusion model (Karras et al., 2022) for this purpose. The time schedule adheres to the recommended setting with $\rho = 7$. With a fixed level of stochasticity, parameterized by $\eta$, we contrast the second order Heun method with our method at NFE values of $6, 8, 10, 15, 20, 25, 30, 35, 50, 75, 100, 150, 200, 350, 500$. We also examine the influence of $\eta$ on the FID with a fixed NFE of 75, iterating $\eta = 0, 0.01, 0.025, 0.05, 0.15, 0.20, 0.30, 0.35, 0.40, 0.45, 0.50$. We observe that adding random noise can enhance perceptual quality in terms of FID when compared to deterministic sampling, although an excess of stochasticity can impair sampling quality. We use the official code from Karras et al. (2022) to calculate FID score.

### F.5 Reproduction of Fig 4b

This experiment is designed to corroborate the effectiveness of our enhanced order analysis and RES in large-scale text-to-image diffusion models (DeepFloyd, 2023), primarily against the benchmark set by DPM-Solver++ (Lu et al., 2022b). The initial experiments examine FID-CLIP scores under varying classifier-free guidance weights (Ho & Salimans, 2022). With classifier-free guidance, we can build a new diffusion model based on a conditional diffusion model and an unconditional model. Specifically, for either the data prediction model or the noise prediction model, denoted by their network as $g$, the newly proposed diffusion model adheres to the following protocol:

$$\hat{g}(\boldsymbol{x}, \sigma, \boldsymbol{c}) := g(\boldsymbol{x}, \sigma, \emptyset) + \omega(g(\boldsymbol{x}, \sigma, \boldsymbol{c}) - g(\boldsymbol{x}, \sigma, \emptyset)), \tag{52}$$

where $\omega$ serves as the guidance weight, $\emptyset$ represents the unconditional signal, and $\boldsymbol{c}$ stands for the conditional signal[1]. We vary the guidance between $1, 1.5, 3, 5, 10$. We employ ViT-g-14 (Ilharco et al., 2021) for the calculation of the CLIP score and pytorch-fid for FID. This experiment is carried out on the MS-COCO validation dataset. Owing to computational resource constraints, the experiment is performed at a resolution of $64 \times 64$ with a fixed 25 NFE. It should be noted that for this experiment, we merely substitute the degenerated Butcher tableau of DPM-Solver++ with our Butcher tableau, keeping all other configurations consistent. The experiment is based on IF-I-XL.

In a further experiment, we compare RES and DPM-Solver++ using the upsampling diffusion model IF-II-L, which upsamples a low-resolution image of $64 \times 64$ pixels to a high-resolution image of $256 \times 256$ pixels. For this test, we feed generated $64 \times 64$ images, using the single-step RES with 49 NFE, along with corresponding captions into the upsampling model. We then vary the NFE to assess the quality of the generated images. Intriguingly, our findings reveal that the single-step RES outperforms multi-step methods in terms of the quality of samples produced.

In our experiments involving DeepFloyd IF models, we adopt the time schedule outlined in Eq (48) with $\rho = 7$. Owing to constraints on our computational resources, we are unable to sweep through hyperparameters to locate a more optimized time schedule. However, we posit that with the same NFE, the quality of the sampling could be further enhanced by fine-tuning the time schedule.

### F.6 Licenses

**Dataset**

- ImageNet (Russakovsky et al., 2015)                    The license status is unclear
- MS-COCO (Lin et al., 2014)                    Creative Commons Attribution 4.0 License.

**Pre-trained models and code**

- EDM ImageNet (Karras et al., 2022)                    Creative Commons Attribution-NonCommercial-ShareAlike 4.0 International License
- DeepFloyd IF (DeepFloyd, 2023)                    DeepFloyd IF License Agreement
- OpenAI Guided ImageNet (Dhariwal & Nichol, 2021)    MIT License
- Pytorch FID (Seitzer, 2020)                    Apache License 2.0
- OpenCLIP (Ilharco et al., 2021)                    MIT License

## G   Image samples

In this section, we include samples from our experiments for quantitative comparison.

---

[1]We slightly modify the classifier-free equation to align it with the implementation (DeepFloyd, 2023) available at `https://github.com/deep-floyd/IF/blob/2ad4dff7cde3cce91f237270a3cc81cae4578015/deepfloyd_if/modules/base.py#L114`

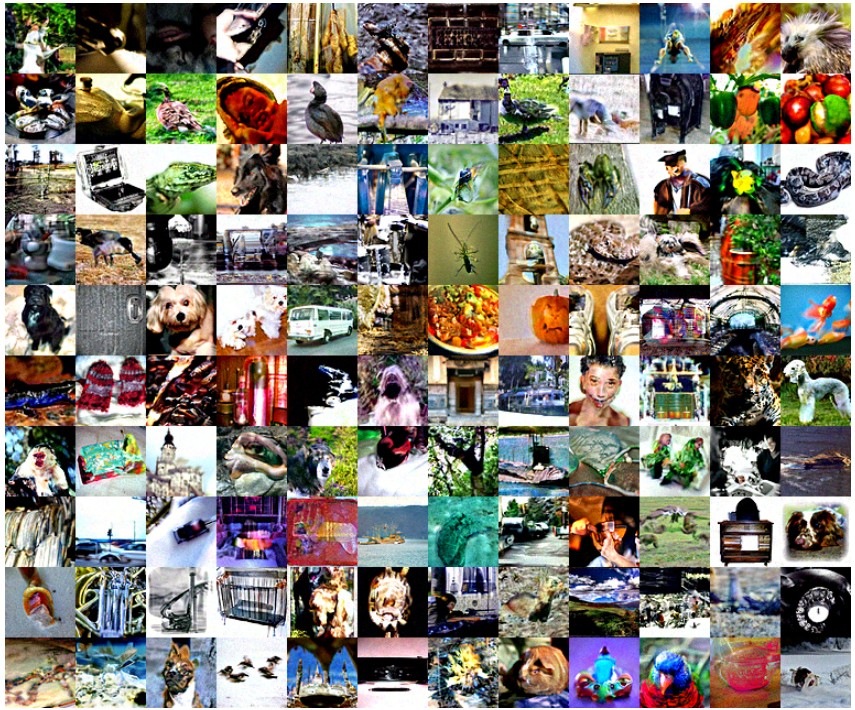

**Figure 7:** Uncurated samples of $64 \times 64$ ImageNet model (Karras et al., 2022) with single-step Heun 9 NFE. (FID=35.31)

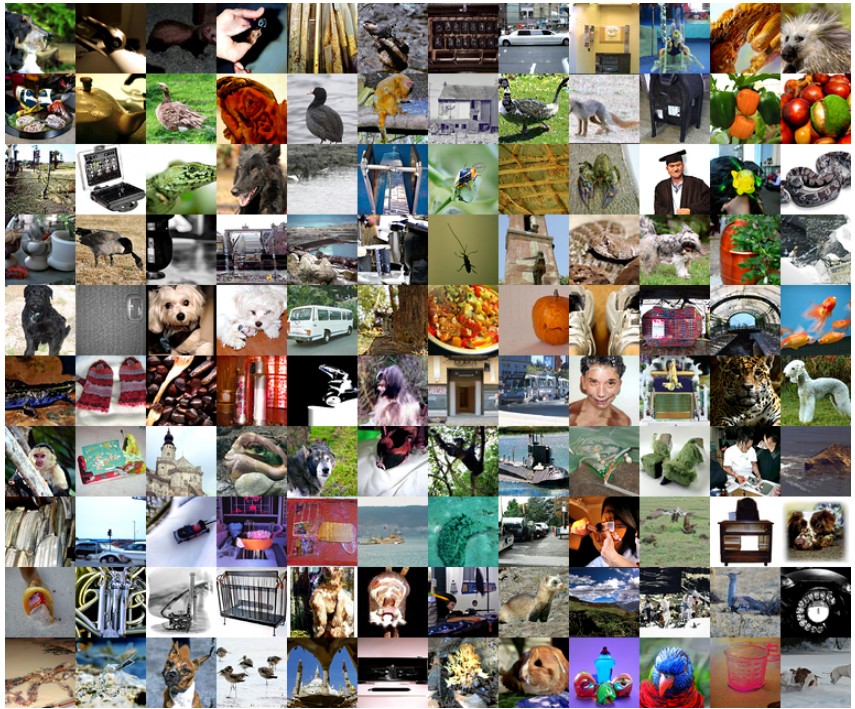

**Figure 8:** Uncurated samples of $64 \times 64$ ImageNet model (Karras et al., 2022) with single-step Heun 15 NFE. (FID=5.54)

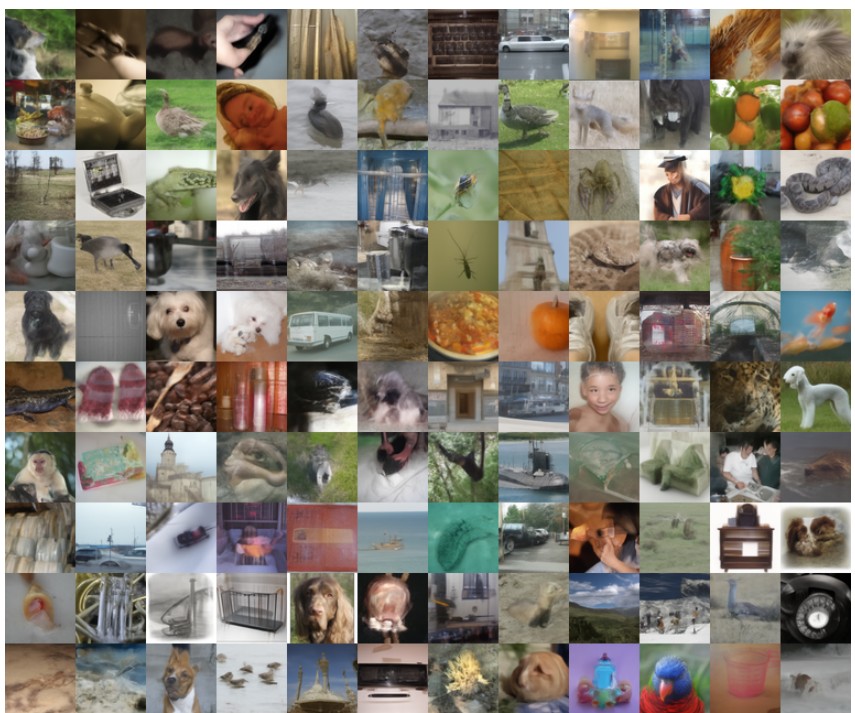

**Figure 9:** Uncurated samples of $64 \times 64$ ImageNet model (Karras et al., 2022) with single-step DPM-Solver++ 9 NFE. (FID=16.77)

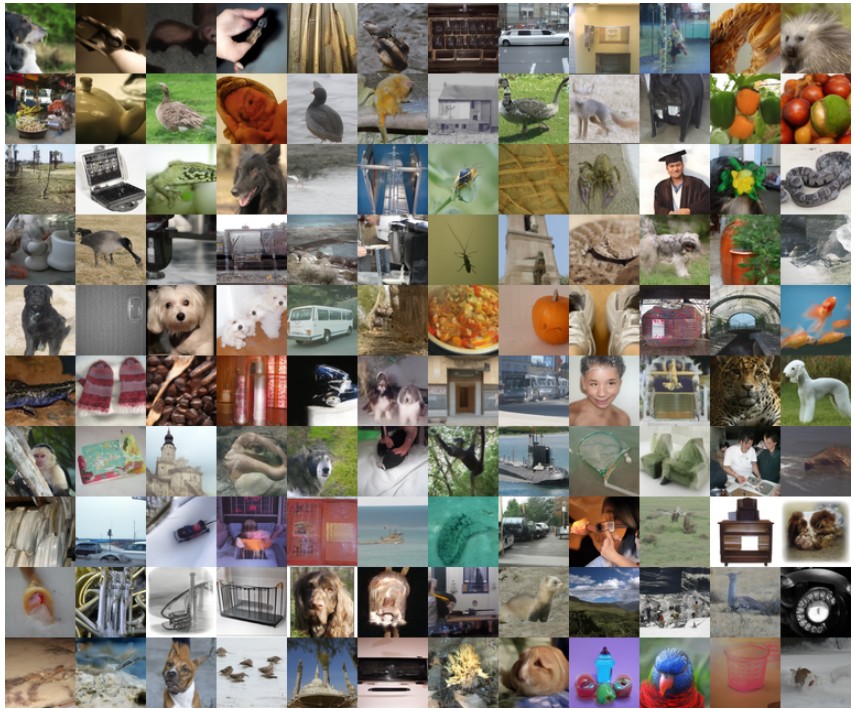

**Figure 10:** Uncurated samples of $64 \times 64$ ImageNet model (Karras et al., 2022) with single-step DPM-Solver++ 15 NFE. (FID=6.40)

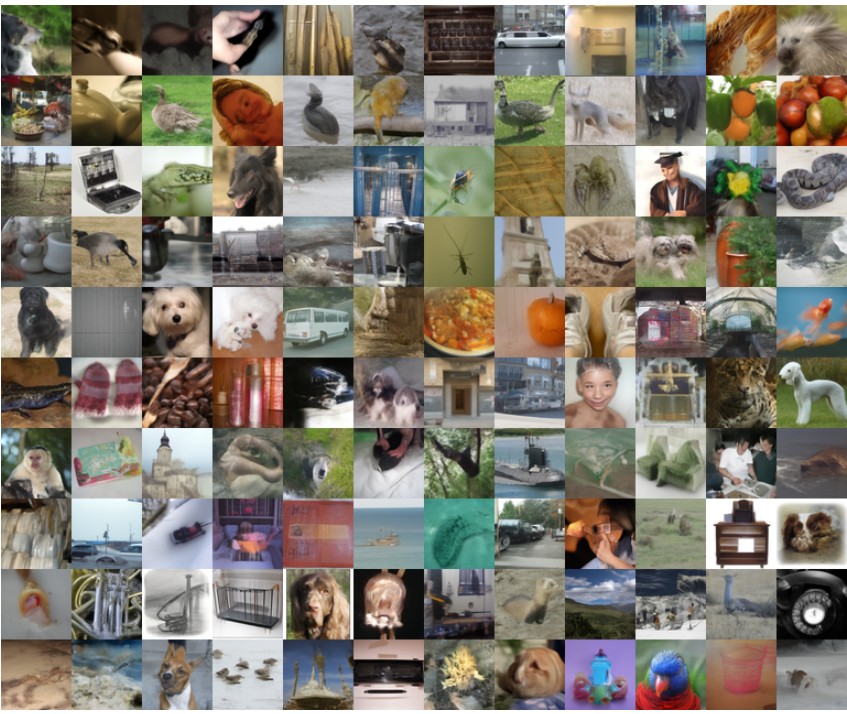

**Figure 11:** Uncurated samples of $64 \times 64$ ImageNet model (Karras et al., 2022) with single-step RES data prediction model 9 NFE. (FID=8.66)

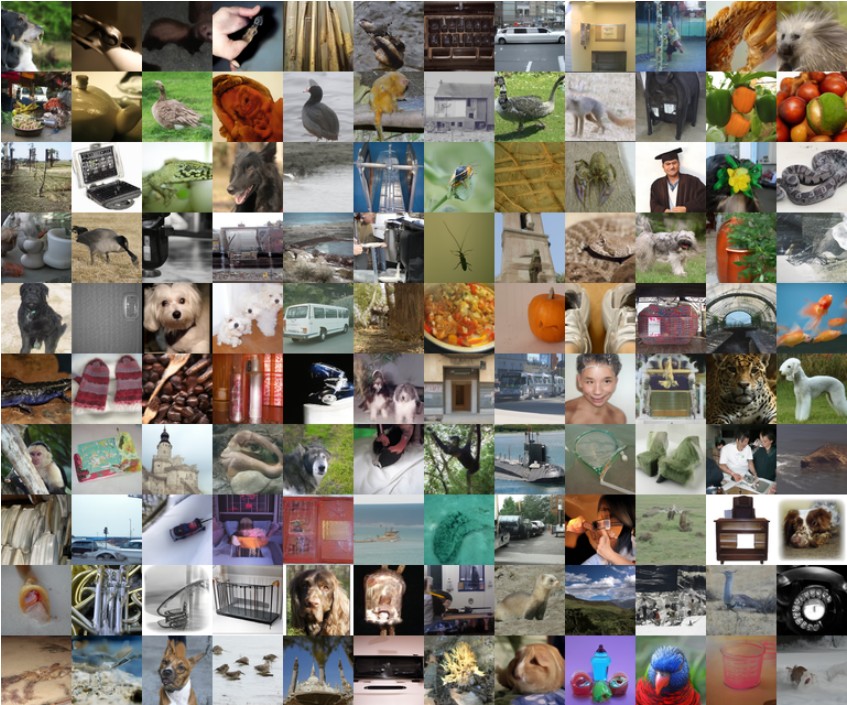

**Figure 12:** Uncurated samples of $64 \times 64$ ImageNet model (Karras et al., 2022) with single-step RES data prediction model 15 NFE. (FID=3.92)

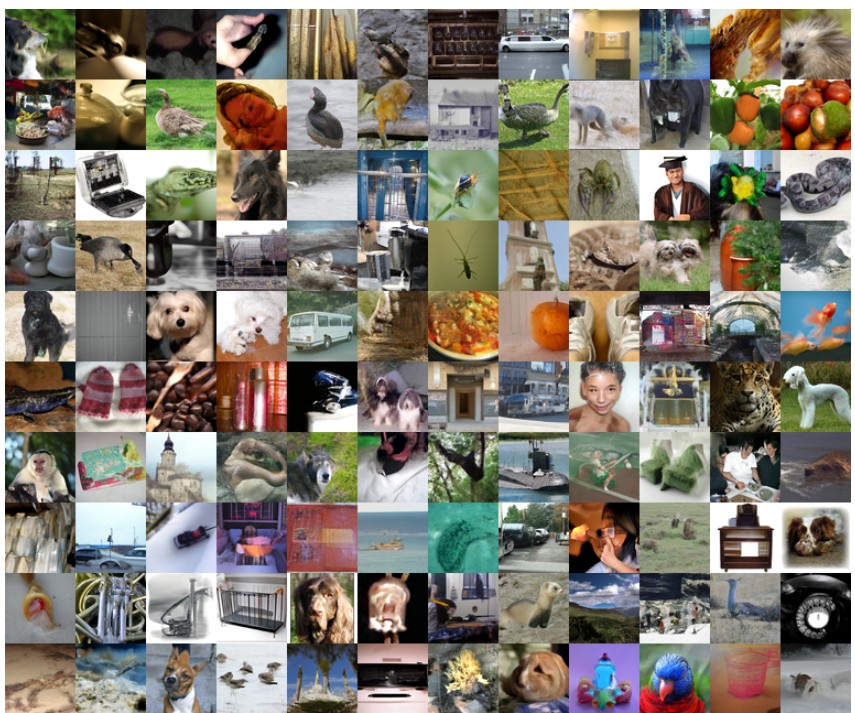

**Figure 13:** Uncurated samples of $64 \times 64$ ImageNet model (Karras et al., 2022) with single-step RES noise prediction model 9 NFE. (FID=5.08)

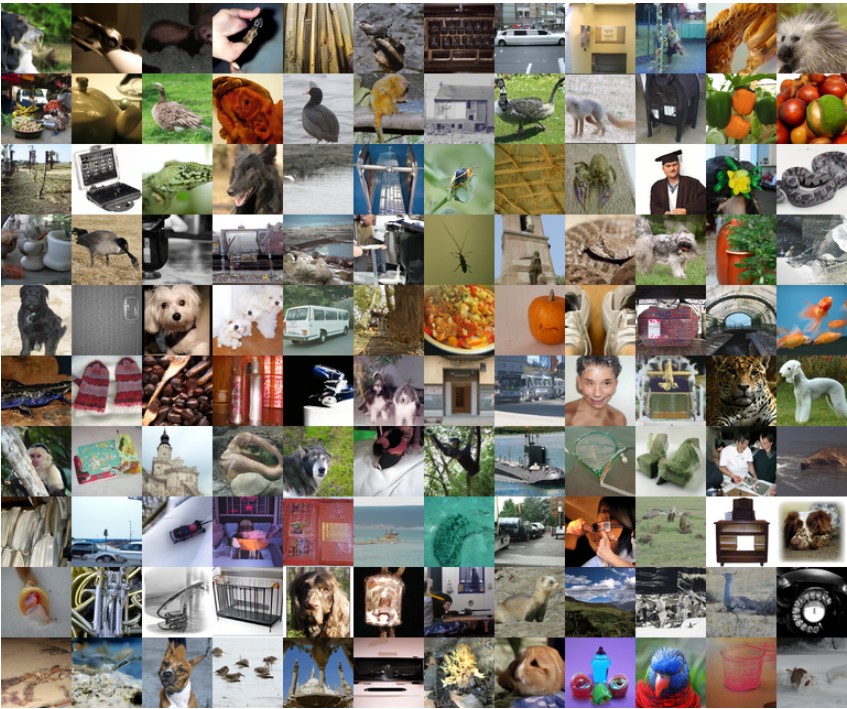

**Figure 14:** Uncurated samples of $64 \times 64$ ImageNet model (Karras et al., 2022) with single-step RES noise prediction model 15 NFE. (FID=3.88)

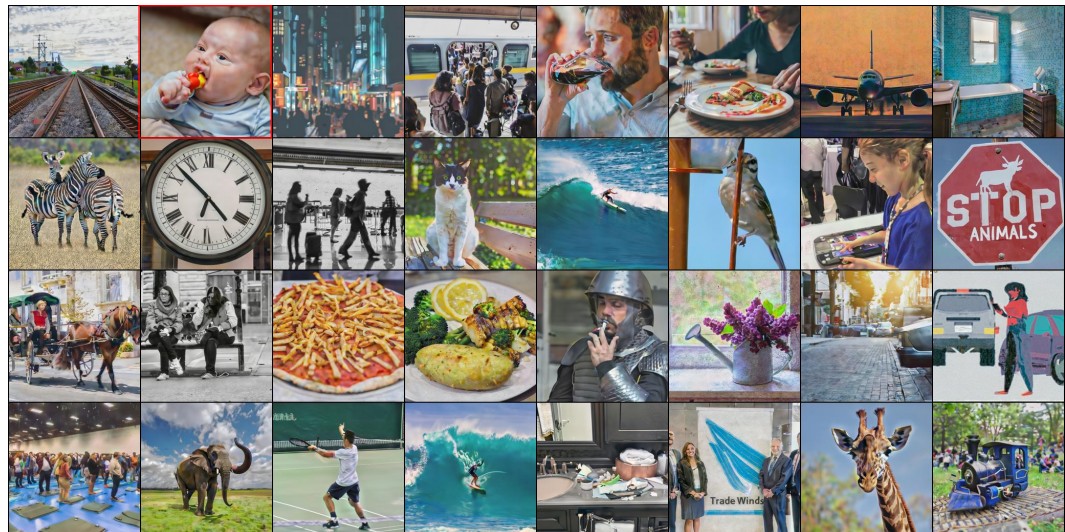

Uncurated samples of $256 \times 256$ DeepFloyd IF model with single-step DPM++ 19 NFE  (MS-COCO validation FID=24.57)

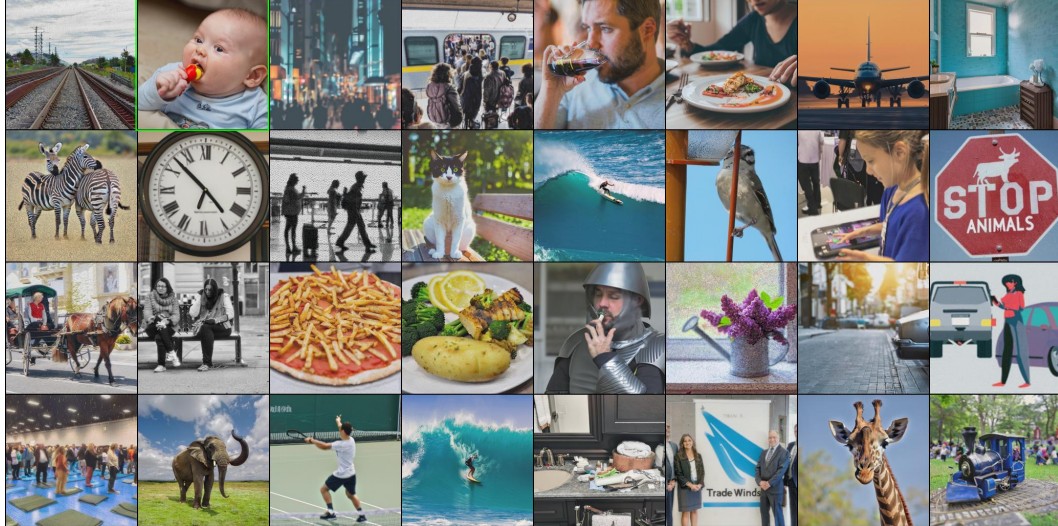

Uncurated samples of $256 \times 256$ DeepFloyd IF model with single-step RES19 NFE  (MS-COCO validation FID=24.21)

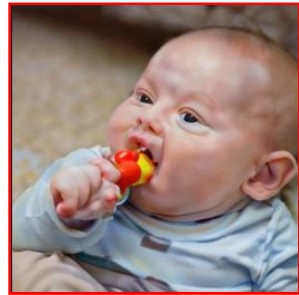 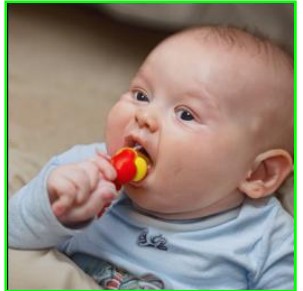

**Figure 15:** Single-step DPM-Solver++ *vs* Single-step RES on cascaded DM

