# OpenReview forum: "Improved order analysis and design of exponential integrator for diffusion models sampling"
_ICLR.cc/2024/Conference — Submitted to ICLR 2024_

### Official Review · Reviewer_XmiM · 2023-10-20

**Soundness:** 3 good
**Presentation:** 2 fair
**Contribution:** 1 poor
**Rating:** 3
**Confidence:** 4

**Summary:**

Sampling in diffusion models requires solving the "probability flow" ordinary differential equation (ODE).
The submission discusses different parametrisations of the probability flow ODE, which it identifies as a semilinear equation, and argues why some parametrisations may be preferable to others.
Further, the paper discusses order conditions for exponential integrators. It proposes what they coin "refined exponential solver": an exponential integrator that satisfies the prescribed order conditions.
Experiments demonstrate that the proposed solver is more sample-efficient than the DPM-Solver++ (by Lu et al. (2022b)).

**Strengths:**

The paper is well-organised and discusses a significant problem: the choice of numerical solver used in diffusion models, which is crucial because sample efficiency can substantially affect these methods' usability.
The proposed solver consistently outperforms the suggested alternatives, DPM-Solver++ and Heun (but this is unsurprising; see below).

That said, I believe that the strengths of this submission are dominated by a significant weakness:

**Weaknesses:**

I recommend rejection because the paper's main contribution as I understand it, prescribing order conditions for exponential integrators, already exists.
The contributions between Equation (9) and the end of Section 3 and the corresponding proofs in the Appendix reproduce existing derivations without crediting the original work, mainly by Hochbruck and Ostermann (2005).
A more detailed explanation of my assessment follows.

For reference,
I understand that the main contribution of this submission is _deriving_ a set of exponential integrators that satisfy specific order conditions, which makes them provably efficient (when applied to diffusion models).
If I misunderstand the narrative, and this is not the case, it needs to be more clearly stated.
However, if I do understand the narrative correctly:

The Butcher Tableau's of the "refined exponential solver" and the theoretical derivation surrounding the order conditions (Lemma 1, Theorems 1 and 2, and Proposition 1) are known. For example, compare Section 3 to the paper by Hochbruck and Ostermann (2005):

* Lemma 1 is equivalent to Equations 4.9 to 4.12 in the paper by Hochbruck and Ostermann (2005) (in fact, the "proof" in Appendix B.1, introduced as a "key theoretical result" by Appendix B, is an almost verbatim reproduction of Section 4.1 in the mentioned paper but without marking this reproduction as such).
* Theorem 1 is a version of Theorem 4.2 by Hochbruck and Ostermann (2005); Theorem 2 is a version of Theorem 4.3 by Hochbruck and Osterman (2005).
* Proposition 1 paraphrases Theorem 4.7 by Hochbruck and Ostermann (2005).
* The second-order Butcher tableaus in Table 3 are Tables 5.3 and 5.4, and the third-order tableaus are Table 5.9 (RES) in the paper by Hochbruck and Ostermann (2005).
* The observation that the algorithm which the submission called RES outperforms the DPM-solver++ has also been made (and mathematically underpinned) by Hochbruck and Ostermann (2005; Figure 6.3).

See also the survey article by Hochbruck and Ostermann (2010) and the references cited by both works.


In my view, the fact that the analysis already exists reduces the paper's contribution to a point where it should be rejected.
Additionally, I find it problematic that the analysis is presented as new instead of attributing the results to the original works, even though the authors seem to be familiar with the paper by Hochbruck and Ostermann (2005) (the submission cites it on page 18).

**References:**

Hochbruck and Ostermann: Explicit Exponential Runge-Kutta Methods for Semilinear Parabolic Problems. SIAM Journal on Numerical Analysis. 2005.

Hochbruck and Ostermann: Exponential integrators. Acta Numerica. 2010.

**Questions:**

I (mildly) disagree with the assessment in Table 1: EDM/DEIS is also semilinear; I assume the paper means that the linear part in EDM/DEIS is time-varying, whereas the linear parts in logSNR and negative logSNR are time-invariant.

---

> ### Author Response · Authors · 2023-11-16
>
> Thank you for the detailed review and thoughtful feedback. Below we address specific questions.
>
> ## Q1. suggested reference and contributions of our work
> Indeed, our work draws upon well-established theories and analyses from the numerical literature. However, our application of these mathematical tools to diffusion model sampling involves significant efforts and contributions. To the best of our knowledge, these are presented explicitly for the first time in the context of accelerating diffusion models:
> 1. We revisit existing, popular diffusion ODE parameterizations and explicitly detail the transformations that convert probability flow ODEs into canonical semi-linear ODE forms. This includes models for both noise prediction and data prediction.
> 2. Building upon these connections, we utilize analyses and mathematical tools from exponential integrators to conduct error analysis and design superior numerical algorithms.
> 3. We conduct comprehensive experiments to show that our new algorithm, which fully incorporates stratified order conditions, outperforms existing methods in diffusion sampling. The advantage is more obvious when time scheduling is not optimal, see figure 2. Those experiments highlight the importance of designing order-stratified methods.
> 4. Furthermore, extensions based on order-stratified numerical schemes, including both stochastic and multistep methods, also demonstrate superior performance over current methodologies.
>
> ## Q2. analysis is presented as new instead of attributing to the original works.
>
> * Firstly, we acknowledge that key mathematical analysis tools in our study are based on well-established theories in the numerical methods literature. We have updated our revision to emphasize this point more clearly.
> * Secondly, the application of these numerical solvers to diffusion models represents a novel approach. For instance, the utilization of order-satisfying techniques plays a crucial role in accelerating diffusion sampling. Unlike existing works that rely on sophisticated and varied derivations, such as those found in DPM-Solver methods, our approach directly leverages conclusions and numerical methods from the numerical literature. This provides a more unified and mathematically robust explanation and framework.

---

> > ### Author Response · Authors · 2023-11-16
> > **Continue**
> >
> > ## Q3. The work should be rejected because most analyses and algorithms reduce to the one present in a math reference.
> > We strongly contest the reviewer's argument for rejection. Our reasons are as follows:
> > * As highlighted in Q1, only one of our four major steps or contributions is based on the referenced mathematical literature. The remaining three steps represent novel approaches, or at least approaches not yet explored by researchers in diffusion models to the best of our knowledge.
> > * It is important to recognize that works whose main contribution appears to align with algorithms in mathematical references can still offer substantial contributions. Specifically in the context of diffusion models, generative diffusion models, interpreted as a pair of time-reversible forward and backward SDEs[1], are acknowledged in sources like [2,3]. DDIM[4] and DPM-Solvers[5], despite their differing presentations and derivations, essentially distill to the first-order exponential Euler method and the second-order exponential integrator[6] with order defects.
> > * Lastly, the empirical improvements we present are significant. Reducing inference cost is crucial for diffusion models. While some authors or researchers might view our conclusions as predictable, it is the first time that order artifacts in popular and widely used sampling algorithms have been identified, discussed, and shown to substantially accelerate diffusion models through improved order-conditioned algorithms. We do not such results are “unsurprising” to generative AI researcher.
> >
> > We note that the experimental setups, which are low-dimensional synthetical problems, in Hochbruck and Ostermann (2005; Figure 6.3) differ markedly from ours. Besides ODE acceleration, the robustness against various time-scheduling, boosting stochastic, and multistep sampling methods for various diffusion models are never discovered or mentioned to the best of our knowledge.
> >
> > In light of the above discussion, we urge the authors to reconsider the impact and contribution of our work.
> >
> > ## Q4. The assessment in table 1. EDM /DEIS is also semilinear.
> >
> > In our revised submission, we have provided a more detailed classification.
> > When the score model is expressed in the $\epsilon_\theta$ form, the EDM and rho DEIS variants adhere Eq-6. This equation represents a canonical ODE problem, which underpins the use of out-of-the-shelf Runge-Kutta methods in both algorithms.
> > Conversely, if we follow $\mathcal{x}_\theta$ form, resultant Eq-5 assumes a highly nonlinear form, particularly notable where the time variable $\sigma(t)$ appears in the denominator.
> >
> > We have updated our figure and appendix to clarify the distinction more explicitly
> >
> > [1]. Yang Song et al. Score-Based Generative Modeling through Stochastic Differential Equations. ICLR 2021
> >
> > [2]. E Nelson. Dynamical theories of Brownian motion. Press, Princeton, NJ, 1967.
> >
> > [3]. Brian DO Anderson. Reverse-time diffusion equation models. Stochastic Processes and their
> > Applications, 12(3):313–326, 1982
> >
> > [4]. Jiaming Song et al. Denoising Diffusion Implicit Models. ICLR 2021
> >
> > [5]. Cheng Lu et al. DPM-Solver: A Fast ODE Solver for Diffusion Probabilistic Model Sampling in Around 10 Steps. Neurips 2022
> >
> > [6] Marlis Hochbruck et al. Explicit exponential Runge--Kutta methods for semilinear parabolic problems

---

> > > ### Comment · Reviewer_XmiM · 2023-11-20
> > >
> > > Thank you for the reply!
> > >
> > > I read the revised version of the paper. I am happy to see the references to Hochbruck and Ostermann (2005a) and others and acknowledge the other replies to my assessment.
> > >
> > > With the updated version, my concerns are less severe than with the original version; nevertheless, my score remains the same for two reasons:
> > >
> > > 1. The main issue - that the theoretical derivation of order conditions is discussed as being at the centre of this paper - persists. The abstract and introduction need a clear revision, and the introduced "refined exponential solver" remains a known method.
> > > All of these are doable but extensive changes, but I think such a revision is too comprehensive to conduct without resubmitting the manuscript elsewhere.
> > >
> > > 2. As soon as one excludes the theory about the order conditions from the contributions, the remaining ones are too incremental: proposing a different parametrisation of the flow ODE and using a known, "better" exponential integrator (in the sense of order conditions) to solve it would require broader empirical studies than in the paper to be valuable to the ICLR community. (Or, at the very least, the derivations in Section 3.2f could be exchanged with the additional experiments in Appendix F).
> > >
> > >
> > > In both 1. and 2., too much of this paper would need to be rewritten.
> > > Therefore, I maintain my original assessment that this paper should be rejected. Still, I acknowledge the authors' work put into the rebuttals and updating the paper.

---

> ### Author Response · Authors · 2023-11-20
>
> Thanks for reviewer’s response and suggestions.
>
> **Q1:The main issue - that the theoretical derivation of order conditions is discussed as being at the center of this paper - persists.  As soon as one excludes the theory about the order conditions from the contributions, the remaining ones are too incremental**
>
> The central contribution of our paper is the acceleration of the diffusion sampling algorithm, which is built upon a well-established, order-satisfied numerical scheme. We acknowledge that our analysis framework, grounded in the referenced mathematical numeric analysis, may have limited contribution and novelty from the perspective of numerical analysis. However, we emphasize our significant contributions to the field of diffusion models, especially in accelerating visual content generation.
> Notably, our work introduces high-order, order-satisfied diffusion sampling algorithms, previously nonexistent in this domain. We respectfully disagree with the reviewer’s perspective that pointing out prevalent flaws in popular, but order-unsatisfied methods and achieving state-of-the-art diffusion sampling performance is trivial.
> We hope the reviewer can value our contribution to the accelerating diffusion model. Celebrated works such as DDIM and DPM-Solver, while adopting different ways of presenting proposed algorithms, essentially represent special cases that utilize the suboptimal Runge-Kutta tables found in the existing EI literature. Our work situates algorithm development within a more mathematically rigorous framework and conducts a broader range of experiments to demonstrate its effectiveness.
>
> **Q2: broader empirical studies than in the paper to be valuable to the ICLR community**
>
> We believe that our experiments are comprehensive and robust, encompassing a wide range of tests including single-step and multi-step experiments, deterministic and stochastic cases, diffusion models based on labeled and test prompts, and both classifier-based and classifier-free guidance settings. We maintain that our empirical studies are sufficiently broad and valuable to the ICLR community.
>
> **Q3: writing suggestions. abs and introduction. exchange section 3.2 and Appendix F**
>
> We have revised the abstract and introduction to clarify potential misunderstandings.
> Regarding the suggestion to exchange Section 3.2 and Appendix F, we believe that Section 3.2 is pivotal for understanding why current methods like DPM-Solver++ are suboptimal compared to our RES method. Moving this section could impair the readability and logical flow of the paper. Therefore, we prefer to retain the current structure for better coherence and reader engagement.
>
> We appreciate the opportunity to discuss these points and hope that our responses adequately address the concerns raised by the reviewer.

---

### Official Review · Reviewer_xZkC · 2023-11-01

**Soundness:** 3 good
**Presentation:** 2 fair
**Contribution:** 3 good
**Rating:** 6
**Confidence:** 4

**Summary:**

The paper proposes an integration scheme (RES) for the probability flow ODE in diffusion model sampling, which provably attains a higher order of accuracy for the same number of queries to the score estimator.

**Strengths:**

The scheme is clearly extensible to higher order integration methods although the query complexity for such methods will grow. It also provides an interesting framework for understanding other discretizations of the probability flow ODE.

Empirical performance is very clearly better, outperforming the benchmarks by a clear margin on ImageNet. In particular it outperforms the DPM-Solver, and a very clear intuition for this improvement is given (i.e. the order conditions of the latter are not correctly formulated).

**Weaknesses:**

The performance in the secondary experiments (in Figure 4) do not really have a clear message, and in my opinion can be relegated to the appendix.

The performance in the stochastic setting is not so clear cut, and therefore I am inclined to conclude that the primary performance benefits are constrained to the deterministic setting.

In light of the above, I find that this paper makes some clear contributions to the literature on the subject, and I am recommending an accept.

**Questions:**

In Fig 1b, I am not really clear on what the left part of (b) is trying to convey. In my opinion, it is just cluttering up the image and it might be good to remove it.

What is the dashed line in Figure 3?

Can the authors provide more details on the stochastic sampling setting? What is the problem being considered and how can we interpret the results?

Although the uniform Lipschitz assumption has been found in prior works in the literature, it is not particularly satisfactory in light of the processes considered in reality (which are quite degenerate). Can the Assumption be removed from the present work or is it somehow fundamentally required?

Typos:
Final sentence is missing a period (Conclusion)
Appendix A: upsampling -> up sampling

Definition of $\phi_{k+1}$ should be instead $\phi_k$ to match with Eq. 2.11 of Hochbruck.
Satisfy is misspelled in multiple points in the appendix
The Butcher tableau is erroneously called the Buther tableau multiple times in the appendix.
In (43), the product in the exponent should be a sum.

---

> ### Author Response · Authors · 2023-11-16
>
> Thank you for the detailed review and thoughtful feedback. Your comments are helping improve our work. Below we address specific questions.
>
> ## Message from Figure 4
>
> Figure 4 is divided into two parts, each highlighting the significance of implementing order-satisfied numerical schemes in various contexts, including stochastic sampling, text-to-image diffusion, and image super-resolution diffusion:
> - Fig 4(a) presents a comparison between the RES-based stochastic sampling algorithm and the widely-used EDM(Heun)-based stochastic sampling algorithm. This comparison demonstrates that order-satisfied numerical schemes can enhance the efficiency of stochastic sampling.
> - Fig 4(b) explores large-scale text-to-image settings, covering both base text-to-image models and image super-resolution tasks. Our results show that, within the same computational budget, our approaches achieve superior image quality, as indicated by lower FID scores, and better text-image alignment, as evidenced by higher CLIP scores, compared to existing methods. Thanks to great interest in AIGC with text-to-image, we believe RES can accelerate various diffusion applications.
> ## RES for stochastic sampling; what is the setup
>
> Recent studies, including EDM [1], have demonstrated that stochastic sampling often surpasses deterministic methods in performance. Drawing inspiration from stochastic EDM, we adapted our algorithm for a stochastic environment. This involves adding extra stochasticity after each Runge-Kutta (RK) step, which corresponds to a nonzero value of $\eta$  in Algorithm 2. Our approach utilizes the RES single-step update scheme, in contrast to the Heun update-step scheme used by the baseline EDM.
>
> In our study, we explored two key aspects:
>
> - The impact of the number of function evaluations (NFE) on image quality, as measured by the FID score, is shown in the left panel of Figure 4(a).
> - The relationship between NFE and the magnitude of stochasticity (indicated by the magnitude of $\eta$) is presented in the right panel of the same figure.
>
> We have included more discussion on these findings in the revised version of our paper.
>
> ## Figure 1b should be in the appendix
>
> We respectfully disagree with the suggestion to move Figure 1b to the appendix. This figure is central to our analysis, as it illustrates the core methodology for error propagation and the derivation of the order condition. We believe that including this diagram within the main body of the text, particularly in conjunction with Section 3.2, aids readers in grasping the intuition and derivation process more easily.
> ## Can the uniform Lipschitz assumption can be removed from the present work?
>
> To the best of our knowledge, removing the Lipschitz assumption is a non-trivial challenge. As discussed in Section 3.2 and illustrated in Figure 1(b), the primary source of error arises from approximating a non-linear function with a finite number of function evaluations. Without the Lipschitz assumption, the underlying non-linear function could exhibit high oscillations, making it challenging to develop an efficient numerical scheme that effectively minimizes numerical errors in the worst case. We leave the interesting and challenging question for future work.
>
> ## dashline in figure 3
>
> To match the performance of DPM-Solver ++ (S) at 99 NFE, our RES single-step method requires only 59 NFE. This comparison, illustrated in the figure, underscores our method's acceleration capabilities, particularly in the paragraph of "Time-scheduling robustness".
> ## Typos and mistakes in the appendix
> Thanks for the author's detailed review. We fix them in our revision.

---

> ### Author Response · Authors · 2023-11-21
> **A kind reminder for further discussion.**
>
> Thank you again for your review. We hope that we were able to address your concerns in our response. If possible, please let us know if you have any further questions before the reviewer-author discussion period ends. We are glad to address your further concerns, thanks.

---

> > ### Comment · Reviewer_xZkC · 2023-11-21
> > **Response**
> >
> > I thank the authors for their thorough response.  I will elect to keep my original score since I was already recommending acceptance. Nonetheless, I particularly appreciate the added experimental details and I think this will greatly improve the clarity of that section.

---

### Official Review · Reviewer_kkMa · 2023-11-07

**Soundness:** 3 good
**Presentation:** 3 good
**Contribution:** 3 good
**Rating:** 6
**Confidence:** 2

**Summary:**

The paper proposes a novel sampling scheme for diffusion models (DMs) to generate high quality images. In particular, it revisits the probability flow ODE parameterization and proposes a semilinear ODE, which results in RES, a novel sampler for improved error correction. Experiments are conducted on ImageNet with improvement over prior arts.

**Strengths:**

1. The paper is well written and easy to follow.
2. The method shows its effectiveness on ImageNet for unconditional generation.
    - Its convergence on NFE and final quality outperforms DPM-Solver++.
    - Its generated images looks smoother which corresponds to its theoretical settings to have less noises.

**Weaknesses:**

1. Some experiments are worth comparison.
    - For example, with the same NFE, how does it compare to step-distillation method [1].
    - With the same NFE, how does it compare to consistency model [2].
2. Can this method be applied for conditional generation with classifier-free guidance? I don't think the paper mention it.

[1] On Distillation of Guided Diffusion Models.
[2] Consistency Models

**Questions:**

1. In Figure 2, why is the multistep setting even worse than the single step?

---

> ### Author Response · Authors · 2023-11-16
>
> Thank you for the detailed review and thoughtful feedback. Below we address specific questions.
>
> ## How it compared against existing works, such as step-distillation work and consistency models.
>
> * Our work primarily concentrates on training-free methods for pre-trained diffusion models. The majority of our experiments and discussions are centered around methodologies within this category. We have included a discussion of suggested methods in our related work section.
> * The methods you've suggested, such as step-distillation, demand additional training. As for the consistency model, it is indeed a novel approach that offers faster sampling speeds. However, it has not yet been extensively validated in large-scale tasks, such as text-to-image settings, and currently shows a significant performance gap when compared to diffusion models.
> * It's also worth noting that the suggested works, including progressive distillation and consistency distillation, rely on ODE solvers as a foundational element or 'teacher'. Therefore, our method has the potential to accelerate these processes as well.
> ## Can this method be applied for conditional generation with classifier-free guidance? I don't think the paper mention it.
> Our paper indeed addresses the application of our method in various settings, including conditional generation with classifier-free guidance:
> * In our text-to-image (DeepFloyd IF) experiments, both Figure 2 and Figure 4 in the paper illustrate how our approach accelerates classifier-free guidance.
> * Additionally, for classifier-based guidance, we have included relevant experiments in the appendix. Please refer to Figure 6 for detailed insights.
>
> ## In fig2, why is the multistep setting even worse than the single step?
>
> We believe this question pertains to the third subplot in Figure 3, where various methods are tested under a suboptimal time schedule. Both multistep and single-step methods approximate the nonlinear function using a polynomial. However, unlike single-step methods that query new function evaluations, multistep methods attempt to save on evaluations by reusing previous ones, which may become outdated. In a suboptimal time schedule, the disadvantage of relying on these outdated evaluations could negate the benefit of conserving new evaluations. This results in poorer performance compared to single-step methods, which, despite incurring extra evaluations, obtain up-to-date information for nonlinear function approximation.

---

> ### Author Response · Authors · 2023-11-21
> **A kind reminder for further discussion.**
>
> Thank you again for your review. We hope that we were able to address your concerns in our response. If possible, please let us know if you have any further questions before the reviewer-author discussion period ends. We are glad to address your further concerns, thanks.

---

### Author Response · Authors · 2023-11-16
**Update and response to AC/Reviewers**

We would like to express our gratitude for AC and all reviewers for their insightful comments and feedback. We are glad to find that all reviewers appreciate the **theoretical importance of order-satisfied numerical solvers** and the **significant improvement of our method over baseline methods in various empirical experiments**. Thanks to the importance of accelerating diffusion sampling for AIGC applications and the encouraging results we achieved, we are confident that our algorithm can reduce the inference cost of various generative AI applications.

We have updated our manuscript to address the concerns raised by Reviewers xZkc and XmiM. Reviewer xZkc pointed out the need for more comprehensive explanations of our experimental results in the main paper; adjustments have been made accordingly to minimize confusion. Reviewer XmiM raised issues regarding missing references and the originality of some theoretical results. In response, we have emphasized in our revision that we are built upon well-established numerical analysis. We also have highlighted that it is only after providing a unified framework, resulting in a well-studied semi-linear ODE, that we are able to leverage the above math tools. Although ODE order analysis is a known field for numerical experts, we highlight that researchers in generative modeling often overlook the significance of developing order-satisfied methods. Surprisingly, these methods can have a substantial impact on diffusion model sampling, yet many continue to use sub-optimal algorithms.

---

### Meta-Review · Area_Chair_gMRE · 2023-12-10

**Metareview:**

This paper focuses on advances in differential equation solvers for diffusion models (DMs), particularly focusing on exponential integrators (EIs). The authors focus their attention on an issue which is that despite their effectiveness, current high-order EI-based sampling algorithms suffer from reliance on degenerate EI solvers, leading to poorer error bounds and accuracy. They claim that this makes sampling quality sensitive to design elements like time-step schedules. To address these issues, the paper revisits the design of differential solvers for DMs and introduces Refined Exponential Solver (RES). The authors claim that this overcomes the limitations of existing high-order EI solvers by fulfilling essential order conditions. They also purport that this results in better theoretical error bounds and practical improvements in sampling efficiency and stability. The authors also carryout experiments suggesting performance enhancements in a pre-trained ImageNet diffusion model when switching from the DPM-Solver++ to the RES in the sense of decrease in numerical defects and improvement in FID scores.

The reviewers thought the paper is well written and the experiments are nice. They however raised a variety of concerns about comparison with other related literature such as stop-distillation and consistency models, not clear performance benefit in the stochastic setting. Furthermore one reviewer raised serious concerns about the contributions of this paper and lack of reference to existing work which was used quite strongly. Some of the above concerns were alleviated and the authors did cite and explain this prior work better. However, the reviewer was still not fully satisfied about the novelty of the paper. IMO the paper has interesting ideas and nice experiments but after a closer read to me also the contributions w.r.t. existing literature needs to be clarified. It is clear to me that this paper needs to go through another meticulous review for a second time therefore I recommend rejection but encourage the authors to resubmit to a future venue after addressing the concerns of the reviewer more carefully.

**Justification For Why Not Higher Score:**

The authors had a spirited discussion with the negative reviewer. While some concerns were alleviated it is clear that still there are issues that need to be addressed.

**Justification For Why Not Lower Score:**

N/A

---

### Decision · Program_Chairs · 2024-01-16

Reject